# CRYSTALFORMER: INFINITELY CONNECTED ATTENTION FOR PERIODIC STRUCTURE ENCODING

**Tatsunori Taniai[1], Ryo Igarashi[1], Yuta Suzuki[2], Naoya Chiba[3], Kotaro Saito[4,5]**
**Yoshitaka Ushiku[1], Kanta Ono[5]**
[1]OMRON SINIC X Corp., [2]Toyota Motor Corp., [3]Tohoku University, [4]Randeft Inc., [5]Osaka University
`https://omron-sinicx.github.io/crystalformer/`

## ABSTRACT

Predicting physical properties of materials from their crystal structures is a fundamental problem in materials science. In peripheral areas such as the prediction of molecular properties, fully connected attention networks have been shown to be successful. However, unlike these finite atom arrangements, crystal structures are infinitely repeating, periodic arrangements of atoms, whose fully connected attention results in *infinitely connected attention*. In this work, we show that this infinitely connected attention can lead to a computationally tractable formulation, interpreted as *neural potential summation*, that performs infinite interatomic potential summations in a deeply learned feature space. We then propose a simple yet effective Transformer-based encoder architecture for crystal structures called *Crystalformer*. Compared to an existing Transformer-based model, the proposed model requires only 29.4% of the number of parameters, with minimal modifications to the original Transformer architecture. Despite the architectural simplicity, the proposed method outperforms state-of-the-art methods for various property regression tasks on the Materials Project and JARVIS-DFT datasets.

## 1 INTRODUCTION

Predicting physical properties of materials from their crystal structures without actually synthesizing materials is important for accelerating the discovery of new materials with desired properties (Pollice et al., 2021). While physical simulation methods such as density functional theory (DFT) calculations can accurately simulate such properties, their high computational load largely limits their applicability, for example, in large-scale screening of potentially valuable materials. Thus, high-throughput machine learning (ML)-based approaches are actively studied (Pollice et al., 2021; Choudhary et al., 2022).

Since crystal structures and molecules are both 3D arrangements of atoms, they share similar challenges in their encoding for property prediction, such as permutation invariance and SE(3) invariance (i.e., rotation and translation invariance). Hence, similar approaches using graph neural networks (GNNs) are popular for invariantly encoding these 3D structures (Reiser et al., 2022).

Against this mainstream of GNN variants, approaches based on Transformer encoders (Vaswani et al., 2017) are emerging recently and showing superior performance in property prediction of molecules (Ying et al., 2021; Liao & Smidt, 2023) and crystal structures (Yan et al., 2022) . Particularly, Graphormer by Ying et al. (2021) adopts fully connected attention between atoms in a molecule, by following the standard Transformer architecture, and showed excellent prediction performance.

Crystal structures, however, have a unique structural feature —periodicity— that produces infinitely repeating periodic arrangements of atoms in 3D space. Because of the periodicity, fully connected attention between atoms in a crystal structure leads to a non-trivial formulation, namely *infinitely connected attention* (see Fig. 1), which involves infinite series over repeated atoms. The previous method, called Matformer (Yan et al., 2022), avoids such a formulation and presents itself rather as a hybrid of Transformer and message passing GNN. Thus, whether the standard Transformer architecture is effectively applicable to crystal structure encoding is still an open question.

In this work, we interpret this infinitely connected attention as a physics-inspired infinite summation of interatomic potentials performed deeply in abstract feature space, which we call *neural potential*

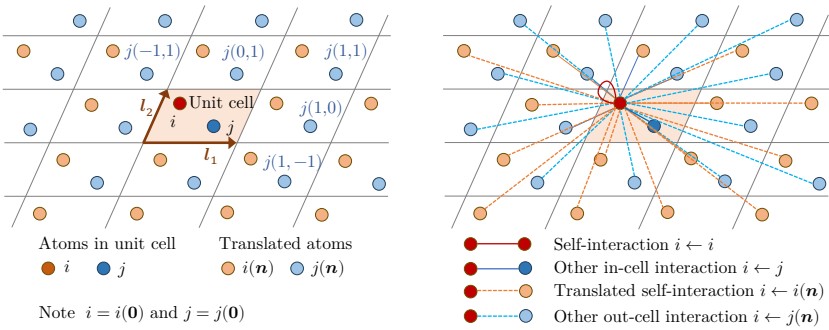

Figure 1: **2D diagrams of crystal structure and infinitely connected attention.**

*summation*. In this view, attention weights are formulated as interatomic distance-decay potentials, which make the infinitely connected attention approximately tractable. By using this formulation, we propose a simple yet effective Transformer encoder for crystal structures called *Crystalformer*, establishing a novel Transformer framework for periodic structure encoding. Compared to the previous work by Yan et al. (2022), we aim to develop the formalism for more faithful Transformer-based crystal structure encoding. The resulting architecture is shown to require only 29.4% of the total number of parameters of their Matformer to achieve better performance, while sharing useful invariance properties. We also point out that a direct extension of Graphormer (Ying et al., 2021) for periodic structures leads to inadequate modeling of periodicity, which the proposed framework overcomes. We further show that the proposed framework using the infinitely connected attention formulation is beneficial for efficiently incorporating long-range interatomic interactions.

Quantitative comparisons using Materials Project and JARVIS-DFT datasets show that the proposed method outperforms several neural-network-based state-of-the-art methods (Xie & Grossman, 2018; Schütt et al., 2018; Chen et al., 2019; Louis et al., 2020; Chen & Ong, 2022; Choudhary & DeCost, 2021; Yan et al., 2022) for various crystal property prediction tasks. We release our code online.

## 2 PRELIMINARIES

We begin with introducing the unit cell representation of crystal structures and recapping standard self-attention in Transformer encoders with relative position representations.

### 2.1 CRYSTAL STRUCTURES

Assume a periodic structure system on a lattice in 3D space, which consists of finite sets of points and their attributes, $\mathcal{P} = \{\boldsymbol{p}_1, \boldsymbol{p}_2, ..., \boldsymbol{p}_N\}$ and $\mathcal{X}^0 = \{\boldsymbol{x}_1^0, \boldsymbol{x}_2^0, ..., \boldsymbol{x}_N^0\}$, in a unit cell as well as lattice vectors, $\boldsymbol{l}_1, \boldsymbol{l}_2, \boldsymbol{l}_3 \in \mathbb{R}^3$, defining the unit cell translations. As a crystal structure, each $\boldsymbol{p}_i \in \mathbb{R}^3$ and $\boldsymbol{x}_i^0 \in \mathbb{R}^d$ represent the Cartesian coordinate and species of an atom in the unit cell, respectively. Any point in the system is then located at a translated position, or an *image*, of a unit-cell point as

$$\boldsymbol{p}_{i(\boldsymbol{n})} = \boldsymbol{p}_i + n_1 \boldsymbol{l}_1 + n_2 \boldsymbol{l}_2 + n_3 \boldsymbol{l}_3, \tag{1}$$

where three integers in $\boldsymbol{n} = (n_1, n_2, n_3) \in \mathbb{Z}^3$ define a 3D translation with $\boldsymbol{l}_1, \boldsymbol{l}_2, \boldsymbol{l}_3$. See Fig. 1 (left) for an illustration. For brevity, let $i$ denote the index of the $i$-th unit-cell point without translation, and let $i(\boldsymbol{n})$ denote the index of a translated image of $i$ (including $i$ itself as $i(\boldsymbol{0})$). $\sum_{\boldsymbol{n}}$ denotes the infinite series over $\mathbb{Z}^3$. Indices $j$ and $j(\boldsymbol{n})$ are used similarly.

### 2.2 SELF-ATTENTION WITH RELATIVE POSITIONS

Self-attention (Vaswani et al., 2017) with relative position representations (Shaw et al., 2018) transforms an ordered finite sequence of feature vectors to another, as $(\boldsymbol{x}_1, ..., \boldsymbol{x}_N) \rightarrow (\boldsymbol{y}_1, ..., \boldsymbol{y}_N)$:

$$\boldsymbol{y}_i = \frac{1}{Z_i} \sum_{j=1}^{N} \exp\left(\boldsymbol{q}_i^T \boldsymbol{k}_j / \sqrt{d_K} + \phi_{ij}\right)\left(\boldsymbol{v}_j + \boldsymbol{\psi}_{ij}\right), \tag{2}$$

where query $\boldsymbol{q}$, key $\boldsymbol{k}$, value $\boldsymbol{v}$ are linear projections of input $\boldsymbol{x}$, $Z_i = \sum_{j=1}^N \exp(\boldsymbol{q}_i^T \boldsymbol{k}_j / \sqrt{d_K} + \phi_{ij})$ is the normalizer of softmax attention weights, and $d_K$ is the dimensionality of $\boldsymbol{k}$ and $\boldsymbol{q}$. Since each input $\boldsymbol{x}_i$ is position-agnostic and so are pairwise similarity $\boldsymbol{q}_i^T \boldsymbol{k}_j / \sqrt{d_K}$ and value $\boldsymbol{v}_j$, they are augmented with scalar $\phi_{ij}$ and vector $\boldsymbol{\psi}_{ij}$ biases that encode/embed the relative position, $j - i$.

## 3 CRYSTALFORMER

We consider a problem of estimating physical properties of a given crystal structure. Following Xie & Grossman (2018) and Schütt et al. (2018), we represent a physical state of the whole structure as a finite set of abstract state variables for the atoms in the unit cell, $\mathcal{X} = \{\boldsymbol{x}_1, \boldsymbol{x}_2, ..., \boldsymbol{x}_N\}$, assuming that any atom in the structure shares the state with its corresponding unit-cell atom, as $\boldsymbol{x}_{i(\boldsymbol{n})} = \boldsymbol{x}_i$ . Given input state $\mathcal{X}^0$ that only symbolically represents the species of the unit-cell atoms, we evolve state $\mathcal{X}^0$, through repeated interactions between atom-wise states, to another $\mathcal{X}'$ reflecting the target properties for prediction. To this end, we propose an attention operation for state $\mathcal{X}$, which is induced by the input structure specified with the unit cell points $\{\boldsymbol{p}_1, \boldsymbol{p}_2, ..., \boldsymbol{p}_N\}$ and lattice vectors $\{\boldsymbol{l}_1, \boldsymbol{l}_2, \boldsymbol{l}_3\}$.

### 3.1 INFINITELY CONNECTED ATTENTION AS NEURAL POTENTIAL SUMMATION

Inspired by Graphormer (Ying et al., 2021)'s fully connected attention for atoms in a molecule, we formulate similar dense attention for a crystal structure for its state evolution. Compared to finite graph representations in GNNs, this dense formulation more faithfully represents the physical phenomena occurring inside crystal structures and should provide a good starting point for discussion. Because of the crystal periodicity, such attention amounts to pairwise interactions between the unit-cell atoms $i$ as queries and all the infinitely repeated atoms $j(\boldsymbol{n})$ in the structure as keys/values as

$$\boldsymbol{y}_i = \frac{1}{Z_i} \sum_{j=1}^N \sum_{\boldsymbol{n}} \exp\left(\boldsymbol{q}_i^T \boldsymbol{k}_{j(\boldsymbol{n})} / \sqrt{d_K} + \phi_{ij(\boldsymbol{n})}\right) \left(\boldsymbol{v}_{j(\boldsymbol{n})} + \boldsymbol{\psi}_{ij(\boldsymbol{n})}\right), \tag{3}$$

where $Z_i = \sum_{j=1}^N \sum_{\boldsymbol{n}} \exp(\boldsymbol{q}_i^T \boldsymbol{k}_{j(\boldsymbol{n})} / \sqrt{d_K} + \phi_{ij(\boldsymbol{n})})$. Notice $\boldsymbol{k}_{j(\boldsymbol{n})} = \boldsymbol{k}_j$ and $\boldsymbol{v}_{j(\boldsymbol{n})} = \boldsymbol{v}_j$ since $\boldsymbol{x}_{j(\boldsymbol{n})} = \boldsymbol{x}_j$. Fig. 1 (right) illustrates these dense connections. We regard this infinitely connected attention in Eq. (3) as a *neural potential summation*. In physical simulations, energy calculations typically involve infinite summation $\sum_{j(\boldsymbol{n}) \neq i} \Phi(\|\boldsymbol{p}_{j(\boldsymbol{n})} - \boldsymbol{p}_i\|) v_{j(\boldsymbol{n})}$ for potential function $\Phi(r)$ and physical quantity $v$ (*e.g.*, electric charge). Analogously, Eq. (3) computes the state of each unit-cell atom, $\boldsymbol{y}_i$, by summing abstract influences, $(\boldsymbol{v}_{j(\boldsymbol{n})} + \boldsymbol{\psi}_{ij(\boldsymbol{n})})$, from all the atoms in the structure, $j(\boldsymbol{n})$, with their weights provided by abstract interatomic scalar potentials, $\exp(\boldsymbol{q}_i^T \boldsymbol{k}_{j(\boldsymbol{n})} / \sqrt{d_K} + \phi_{ij(\boldsymbol{n})})/Z_i$. Since $\boldsymbol{q}_i^T \boldsymbol{k}_{j(\boldsymbol{n})} / \sqrt{d_K}$ and $\boldsymbol{v}_{j(\boldsymbol{n})}$ are position-agnostic, they are augmented with relative position encodings, $\phi_{ij(\boldsymbol{n})}$ and $\boldsymbol{\psi}_{ij(\boldsymbol{n})}$, to reflect the interatomic spatial relation (*i.e.*, $\boldsymbol{p}_{j(\boldsymbol{n})} - \boldsymbol{p}_i$). Thus, $\exp(\phi_{ij(\boldsymbol{n})})$ is interpreted as a spatial dependency factor of the interatomic potential between $i$ and $j(\boldsymbol{n})$, and $\boldsymbol{\psi}_{ij(\boldsymbol{n})}$ as an abstract position-dependent influence on $i$ from $j(\boldsymbol{n})$.

Compared to the standard finite-element self-attention in Eq. (2), the main challenge in computing Eq. (3) is the presence of infinite series $\sum_{\boldsymbol{n}}$ owing to the crystal periodicity.

**Pseudo-finite periodic attention.** With simple algebra (see Appendix A), we can rewrite Eq. (3) as

$$\boldsymbol{y}_i = \frac{1}{Z_i} \sum_{j=1}^N \exp\left(\boldsymbol{q}_i^T \boldsymbol{k}_j / \sqrt{d_K} + \alpha_{ij}\right) \left(\boldsymbol{v}_j + \boldsymbol{\beta}_{ij}\right), \tag{4}$$

where

$$\alpha_{ij} = \log \sum_{\boldsymbol{n}} \exp\left(\phi_{ij(\boldsymbol{n})}\right), \tag{5}$$

$$\boldsymbol{\beta}_{ij} = \frac{1}{Z_{ij}} \sum_{\boldsymbol{n}} \exp\left(\phi_{ij(\boldsymbol{n})}\right) \boldsymbol{\psi}_{ij(\boldsymbol{n})}, \tag{6}$$

and $Z_{ij} = \exp(\alpha_{ij})$. Eq. (4) now resembles the standard finite-element attention in Eq. (2), if $\alpha_{ij}$ and $\boldsymbol{\beta}_{ij}$ are viewed as quantities encoding the relative position between $i$ and $j$ in finite element set $\{1, 2, ..., N\}$. In reality, $\alpha_{ij}$ is the log-sum of spatial dependencies $\exp(\phi_{ij(\boldsymbol{n})})$ between $i$ and the $j$'s

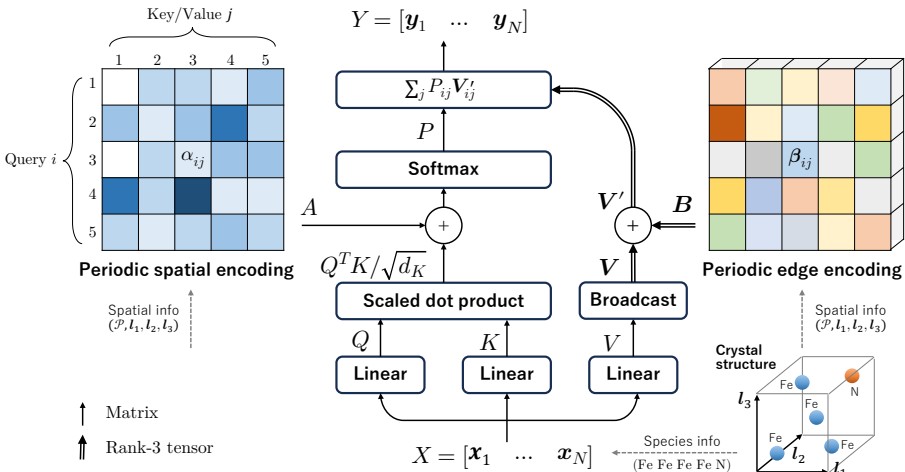

Figure 2: **Pseudo-finite periodic attention with periodic spatial and edge encoding in a matrix-tensor diagram.** Scalar $\alpha_{ij}$ and vector $\boldsymbol{\beta}_{ij}$ integrate the spatial relations between unit-cell atom $i$ and the $j$'s all repeated atoms, allowing the infinitely connected attention to be performed as standard fully connected attention for finite unit-cell atoms. (Unlike usual notation, $X, Q, K, V, Y$ here denote column-vector-based feature matrices for better consistency with the notation in the main text.)

all images. Likewise, $\boldsymbol{\beta}_{ij}$ is the softmax-weighted average of position-dependent influences $\boldsymbol{\psi}_{ij(\boldsymbol{n})}$ on $i$ from the $j$'s all images, weighted by their spatial dependencies. We call $\alpha_{ij}$ and $\boldsymbol{\beta}_{ij}$ the *periodic spatial encoding* and *periodic edge encoding*, respectively, and call the infinitely connected attention with these encodings the *pseudo-finite periodic attention*. If $\alpha_{ij}$ and $\boldsymbol{\beta}_{ij}$ are tractably computed, this attention can be performed similarly to the standard finite-element attention, as shown in Fig. 2.

**Distance decay attention.** In the physical world, spatial dependencies between two atoms tend to decrease as their distance increases. To approximate such dependencies with $\exp(\phi_{ij(\boldsymbol{n})})$, we adopt the following Gaussian distance decay function as a simple choice among other possible forms.

$$\exp(\phi_{ij(\boldsymbol{n})}) = \exp\left(-\|\boldsymbol{p}_{j(\boldsymbol{n})} - \boldsymbol{p}_i\|^2/2\sigma_i^2\right) \tag{7}$$

Here, $\sigma_i > 0$ is a scalar variable controlling the Gaussian tail length. When $\sigma_i$ is not too large, series $\sum_{\boldsymbol{n}}$ in Eqs. (5) and (6) converges quickly as $\|\boldsymbol{n}\|$ grows with a provable error bound (see Appendix B), thus making $\alpha_{ij}$ and $\boldsymbol{\beta}_{ij}$ computationally tractable. To ensure the tractability, we model $\sigma_i$ as a function of current state $\boldsymbol{q}_i$ with a fixed upper bound, as $\sigma_i < \sigma_{\text{ub}}$ (see Appendix C for the complete definition of $\sigma_i$). We found that, although allowing relatively long tails (*e.g.*, $\sigma_{\text{ub}} = 7\text{Å}$) is still tractable, the use of a shorter tail bound (*e.g.*, 2Å) empirically leads to better results. The methodology to eliminate the bound for $\sigma_i$ is further discussed in Sec. 6.

**Value position encoding for periodicity-aware modeling.** Value position encoding $\boldsymbol{\psi}_{ij(\boldsymbol{n})}$ in Eq. (3) represents position-dependent influences on $i$ from $j(\boldsymbol{n})$, and is another key to properly encoding periodic structures. In fact, attention without $\boldsymbol{\psi}_{ij(\boldsymbol{n})}$ cannot distinguish between crystal structures consisting of the same single unit-cell atom with different lattice vectors. This is obvious since Eq. (3) with $\boldsymbol{\psi}_{ij(\boldsymbol{n})} = \boldsymbol{0}$ and $N = 1$ degenerates to $\boldsymbol{y}_1 = \boldsymbol{v}_1$, which is completely insensible to the input lattice vectors (see Appendix D for more discussions). As a simple choice for $\boldsymbol{\psi}_{ij(\boldsymbol{n})}$, we borrow edge features used by existing GNNs (Schütt et al., 2017; Xie & Grossman, 2018). These edge features are Gaussian radial basis functions $\boldsymbol{b}(r) = (b_1, b_2, ..., b_K)^T$ defined as

$$b_k(r) = \exp\left(-(r - \mu_k)^2/2(r_{\max}/K)^2\right), \tag{8}$$

where $\mu_k = kr_{\max}/K$, and $K$ and $r_{\max}$ are hyperparameters. Intuitively, vector $\boldsymbol{b}(r)$ quantizes scalar distance $r$ via soft one-hot encoding using $K$ bins equally spaced between 0 and $r_{\max}$. We provide $\boldsymbol{\psi}_{ij(\boldsymbol{n})}$ as a linear projection of $\boldsymbol{b}(r)$ with trainable weight matrix $W^E$ as

$$\boldsymbol{\psi}_{ij(\boldsymbol{n})} = W^E \boldsymbol{b}\left(\|\boldsymbol{p}_{j(\boldsymbol{n})} - \boldsymbol{p}_i\|\right). \tag{9}$$

**Implementation details.** When computing $\alpha_{ij}$ and $\boldsymbol{\beta}_{ij}$ in Eqs. (5) and (6), the Gaussian functions in series $\sum_{\boldsymbol{n}}$ mostly decays rapidly within a relatively small range of $\|\boldsymbol{n}\|_{\infty} \leq 2$ (*i.e.*, supercell of $5^3$

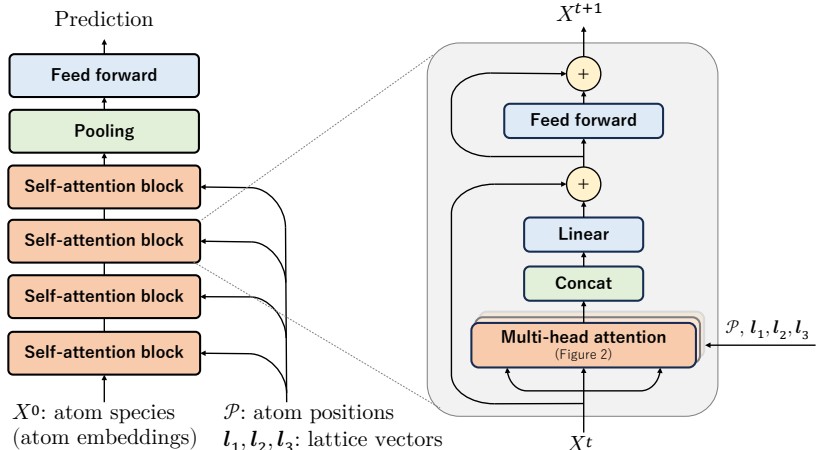

Figure 3: **Network architecture of Crystalformer.**

unit cells), but structures with small unit cells often require larger ranges. We thus adaptively change the range of $n$ for each $i$ to sufficiently cover a radius of $3.5\sigma_i$ in Å. This is done by setting the range of $n_1$ as $-\max(R_1, 2) \le n_1 \le \max(R_1, 2)$ where $R_1 = \lceil 3.5\sigma_i \|l_2 \times l_3\| / \det(l_1, l_2, l_3) \rceil$ ($\lceil \cdot \rceil$ is the ceiling function) and doing similarly for $n_2$ and $n_3$. For Eq. (8) we use $K = 64$ and $r_{\max} = 14$Å.

## 3.2 NETWORK ARCHITECTURE

As illustrated in Fig. 3, the Crystalformer architecture basically follows the original Transformer encoder architecture (Vaswani et al., 2017) with stacked self-attention blocks, each consisting of two residual blocks connecting a multi-head attention (MHA) layer and a shallow feed-forward network (FFN). As an important difference from the original architecture, our self-attention block entirely removes Layer Normalization. We found that a normalization-free architecture with an improved weight initialization strategy proposed by Huang et al. (2020) is beneficial to stabilize the training.

Given a set of trainable embedding vectors (atom embeddings) representing the species of the unit-cell atoms as initial state $\mathcal{X}^0$, Crystalformer transforms it to abstract state $\mathcal{X}'$ through four stacked self-attention blocks using the attention formulation provided in Sec. 3.1. The atom-wise states in $\mathcal{X}'$ are then aggregated into a single vector via the global average pooling. This material-wise feature is further converted through a FFN of linear, ReLU, and final linear layers to predict the target properties. More architectural details are provided in Appendix E.

## 4 RELATED WORK

**Invariant encoders.** Crystal structure encoders for property prediction must satisfy various invariance properties against artificial differences in data representations. The most elementary one is the permutation invariance, which was initially studied by Zaheer et al. (2017) and Qi et al. (2017) and is widely adopted into the GNN framework. Our method is permutation-invariant thanks to the Transformer architecture. The translation and rotation invariance, or the SE(3) invariance, are also essential for ML on 3D point clouds, including molecules and materials. Our method simply employs the fully distance-based modeling (Xie & Grossman, 2018) to ensure the SE(3) invariance. Recent point-cloud encoders, such as convolution-based (Zhang et al., 2019; KIM et al., 2020; Xu et al., 2021) and Transformer-based (Qin et al., 2022; Yu et al., 2023) methods, exploit richer information while maintaining the SE(3) invariance. These techniques can be possibly incorporated to our framework with proper extensions for crystal structures. Lastly, the periodic invariance (*i.e.*, supercell invariance and periodic-boundary shift invariance) has recently been pointed out by Yan et al. (2022) as a property particularly important for crystal structure encoders. Our formalism provides a periodic-invariant encoder, if $\alpha$ and $\beta$ are computed until convergence (see Appendix F).

**GNNs for crystal structure encoding.** The initial successful work on encoding crystal structures with neural networks can be traced back to CGCNN by Xie & Grossman (2018). Their multi-edge distance graph represents crystal structures as finite graphs, and has provided a foundation for the GNN framework for crystal structures. Almost concurrently, Schütt et al. (2018) extended their molecular encoder, SchNet, for crystal structures using a similar approach. Subsequently, several GNNs adopting similar approaches have been proposed for universal property prediction not limited to energy prediction. MEGNet (Chen et al., 2019) proposed to sequentially update atom, bond, and global state attributes through GNN layers. GATGNN (Louis et al., 2020) incorporated attention mechanisms into convolution and pooling layers of the GNN framework. iCGCNN (Park & Wolverton, 2020) and ALIGNN (Choudhary & DeCost, 2021) proposed 3-body interaction GNNs to exploit interatomic angular information, while GeoCGNN (Cheng et al., 2021) used rotation-invariant plane-wave-based edge features to encode directional information. Kosmala et al. (2023) recently proposed Ewald message passing for exploiting long-range interatomic interactions with GNNs. PotNet (Lin et al., 2023) proposed a new type of physics-informed edge feature that embeds the infinite summation value of several known interatomic potentials and allows a standard fully-connected GNN to be informed of crystal periodicity. (We more discuss PotNet in Appendix G.) Apart from material-level property prediction, M3GNet (Chen & Ong, 2022) extended MEGNet for interatomic potential prediction. Our focus is apart from the line of these studies in the GNN framework, and we aim to establish an effective Transformer framework for crystal structure encoding in a standard fashion of the Transformer using fully connected attention.

**Transformers.** The Transformer was originally proposed by Vaswani et al. (2017) for machine translation in natural language processing (NLP), as a sequence-to-sequence model in an encoder-decoder architecture. Since then, it has been widely applied to various tasks in many fields, such as NLP, computer vision, and speech processing (Lin et al., 2022), using its encoder-decoder or encoder-only architecture. Compared to standard convolutional neural network encoders, Transformer encoders have an outstanding ability to model complex interdependencies among input elements (*e.g.*, resolving ambiguous meanings of words in sentence context) and have great flexibility in handling irregularly structured or unordered data such as point clouds (Zeng et al., 2022). These capabilities should benefit the encoding of crystal structures and molecules because atoms in these structures interact with each other in complex ways to determine their states, and also they are structured in 3D space rather than regularly ordered. While there have been attempts to partly replace key modules of GNNs with attention mechanisms, Ying et al. (2021) first presented a complete Transformer architecture (Graphormer) for graph encoding and showed state-of-the-art performance for molecular property prediction. Later, Yan et al. (2022) proposed Matformer as a Transformer-inspired encoder for crystal structures. Matformer fixes the periodic invariance break in existing GNNs by using radius nearest neighbors (radius-NN) instead of k-NN for graph construction. Note that works on language models for materials by Wei et al. (2022) and Fu et al. (2023) are clearly distinguished from ours.

**Graphormer** (Ying et al., 2021) is specifically designed for finite graphs such as molecules and does not ensure distance-decay attention, which leads to its inapplicability to periodic structures. Moreover, even an extended model mending this problem suffers from another modeling problem of periodicity, such as discussed in Sec.3.1. The problem is that Graphormer encodes all the spatial information, including distance edge features, as only softmax biases (*i.e.*, $\phi_{ij}$ in Eq. (2)) and does not use value position encoding (*i.e.*, $\psi_{ij}$). Such a formulation is fine for molecules but fails to distinguish between crystal structures of the same single atom in differently sized unit cells. We will show that this modeling issue leads to performance degradation in Sec. 5.4.

**Matformer** (Yan et al., 2022) employs finite graphs with radius-NN edges similarly to GNNs. Their abstract feature-based attention does not ensure distance decay and is thus inapplicable to the infinitely connected edges. Moreover, they modify the original Transformer in various ways, by extending query/key/value by concatenation as $q_{ij} = (q_i|q_i|q_i), k_{ij} = (k_i|k_j|e_{ij}), v_{ij} = (v_i|v_j|e_{ij})$ ($e_{ij}$ an edge feature), changing the scaled dot product to Hadamard product, softmax to sigmoid function, etc. We consider that the lack of explicit distance decay attention required such modifications, as they report worse results with standard softmax attention. They also propose self-connecting edges (*i.e.*, a part of the orange dashed edges in Fig. 1 (right)) although unsuccessful. By contrast, we employ much denser edges made possible by explicit distance-decay attention. This formulation leads to an architectural framework closely following the original Transformer (see Figs. 2 and 3), specifically its relative-position-based (Shaw et al., 2018) and normalization-free (Huang et al., 2020) variants.

## 5 EXPERIMENTS

We perform regression tasks of several important material properties, comparing with several neural-network-based state-of-the-art methods (Xie & Grossman, 2018; Schütt et al., 2018; Chen et al., 2019; Louis et al., 2020; Chen & Ong, 2022; Choudhary & DeCost, 2021; Yan et al., 2022; Lin et al., 2023). Following our most relevant work by Yan et al. (2022), we use the following datasets with DFT-calculated properties.

**Materials Project (MEGNet)** is a collection of 69,239 materials from the Materials Project database retrieved by Chen et al. (2019). Following Yan et al. (2022), we perform regression tasks of formation energy, bandgap, bulk modulus, and shear modulus.

**JARVIS-DFT (3D 2021)** is a collection of 55,723 materials by Choudhary et al. (2020). Following Yan et al. (2022), we perform regression tasks of formation energy, total energy, bandgap, and energy above hull (E hull). For bandgap, the dataset provides property values obtained by DFT calculation methods using the OptB88vdW functional (OPT) or the Tran-Blaha modified Becke-Johnson potential (MBJ). While MBJ is considered more accurate (Choudhary et al., 2020), we use both for evaluation.

Thanks to the great effort by Choudhary & DeCost (2021) and Yan et al. (2022), many relevant methods are evaluated on these datasets with consistent and reproducible train/validation/test splits. We partly borrow their settings and results for a fair comparison. Doing so also helps to reduce the computational load needed for comparisons.

### 5.1 TRAINING SETTINGS

For each regression task in the Materials Project dataset, we train our model by optimizing the mean absolute error loss function via stochastic gradient descent (SGD) with a batch size of 128 materials for 500 epochs. We initialize the attention layers by following Huang et al. (2020). We use the Adam optimizer (Kingma & Ba, 2015) with weight decay (Loshchilov & Hutter, 2019) of $10^{-5}$ and clip the gradient norm at 1. For the hyperparameters of Adam we follow Huang et al. (2020); we use the initial learning rate $\alpha$ of $5 \times 10^{-4}$ and decay it according to $\alpha\sqrt{4000/(4000 + t)}$ by the number of total training steps $t$, and use $(\beta_1, \beta_2) = (0.9, 0.98)$. For test and validation model selection, we use stochastic weight averaging (SWA) (Izmailov et al., 2018) by averaging the model weights for the last 50 epochs with a fixed learning rate. For the JARVIS-DFT dataset, we use increased batch size and total epochs, as specified in Appendix H, for its relatively smaller dataset size.

### 5.2 CRYSTAL PROPERTY PREDICTION

Tables 1 and 2 summarize the mean absolute errors (MAEs) for totally nine regression tasks of the Materials Project and JARVIS-DFT datasets, comparing our method with eight existing methods [1]. Our method consistently outperforms all but PotNet in all the tasks, even without SWA (*i.e.*, evaluating a model checkpoint with the best validation score). Meanwhile, our method is competitive to PotNet. This is rather remarkable, considering that 1) PotNet exploits several known forms of interatomic potentials to take advantage of a strong inductive bias while we approximate them with simple Gaussian potentials, and that 2) our model is more efficient as shown in the next section. It is also worth noting that our method performs well for the bulk and shear modulus prediction tasks, even with limited training data sizes without any pre-training, which supports the validity of our model.

### 5.3 MODEL EFFICIENCY COMPARISON

Table 3 summarizes the running times [2] and model sizes, comparing with PotNet (Lin et al., 2023) and Matformer (Yan et al., 2022) as the current best GNN-based and Transformer-based models. Notably, the total number of parameters in our model is only 48.6% and 29.4% of the number in

---

[1]Except for M3GNet, their MAEs are cited from previous works (Choudhary & DeCost, 2021; Yan et al., 2022; Lin et al., 2023). We ran M3GNet by using the code provided in Materials Graph Library (`https://github.com/materialsvirtuallab/matgl`).

[2]We evaluated the running times of all the methods by using a single NVIDIA RTX A6000 and a single CPU core on the same machine. We used the codes of PotNet and Matformer in the AIRS OpenMat library (Zhang et al., 2023). The inference time of PotNet is dominated by the precomputation of potential summations (309 ms).

Table 1: **MAE comparison on the Materials Project (MEGNet) dataset.** The three numbers below each property name indicate the sizes of training, validation, and testing subsets. The results of the proposed method without stochastic weight averaging (SWA) are also shown. The numbers in **bold** indicate the best results and the numbers with underline indicate the second best results.

| Method | Formation energy 60000 / 5000 / 4239 eV/atom | Bandgap 60000 / 5000 / 4239 eV | Bulk modulus 4664 / 393 / 393 log(GPa) | Shear modulus 4664 / 392 / 393 log(GPa) |
|---|---|---|---|---|
| CGCNN (Xie & Grossman, 2018) | 0.031 | 0.292 | 0.047 | 0.077 |
| SchNet (Schütt et al., 2018) | 0.033 | 0.345 | 0.066 | 0.099 |
| MEGNet (Chen et al., 2019) | 0.030 | 0.307 | 0.060 | 0.099 |
| GATGNN (Louis et al., 2020) | 0.033 | 0.280 | 0.045 | 0.075 |
| M3GNet Chen & Ong (2022) | 0.024 | 0.247 | 0.050 | 0.087 |
| ALIGNN (Choudhary & DeCost, 2021) | 0.022 | 0.218 | 0.051 | 0.078 |
| Matformer (Yan et al., 2022) | 0.021 | 0.211 | 0.043 | 0.073 |
| PotNet (Lin et al., 2023) | 0.0188 | 0.204 | 0.040 | **0.065** |
| Crystalformer (proposed) | **0.0186** | **0.198** | **0.0377** | 0.0689 |
| Crystalformer w/o SWA | 0.0198 | 0.201 | 0.0399 | 0.0692 |

Table 2: **MAE comparison on the JARVIS-DFT 3D 2021 dataset.** The bandgap prediction is evaluated with the ground truth values provided by two DFT calculation methods, OPT and MBJ.

| Method | Form. energy 44578 / 5572 / 5572 eV/atom | Total energy 44578 / 5572 / 5572 eV/atom | Bandgap (OPT) 44578 / 5572 / 5572 eV | Bandgap (MBJ) 14537 / 1817 / 1817 eV | E hull 44296 / 5537 / 5537 eV |
|---|---|---|---|---|---|
| CGCNN | 0.063 | 0.078 | 0.20 | 0.41 | 0.17 |
| SchNet | 0.045 | 0.047 | 0.19 | 0.43 | 0.14 |
| MEGNet | 0.047 | 0.058 | 0.145 | 0.34 | 0.084 |
| GATGNN | 0.047 | 0.056 | 0.17 | 0.51 | 0.12 |
| M3GNet | 0.039 | 0.041 | 0.145 | 0.362 | 0.095 |
| ALIGNN | 0.0331 | 0.037 | 0.142 | 0.31 | 0.076 |
| Matformer | 0.0325 | 0.035 | 0.137 | 0.30 | 0.064 |
| PotNet | **0.0294** | **0.032** | **0.127** | **0.27** | 0.055 |
| Crystalformer (proposed) | 0.0306 | **0.0320** | 0.128 | 0.274 | **0.0463** |
| Crystalformer w/o SWA | 0.0319 | 0.0342 | 0.131 | 0.275 | 0.0482 |

Table 3: **Efficiency comparison.** We show the average training time per epoch, total training time, and average testing time per material, all evaluated on the JARVIS-DFT formation energy dataset using a single NVIDIA A6000 GPU. We also show the total and number of parameters and number of parameters per self-attention or graph convolution block.

| Model | Type | Time/Epoch | Total | Test/Mater. | # Params. | # Params./Block |
|---|---|---|---|---|---|---|
| PotNet | GNN | 43 s | 5.9 h | 313 ms | 1.8 M | 527 K |
| Matformer | Transformer | 60 s | 8.3 h | 20.4 ms | 2.9 M | 544 K |
| Crystalformer | Transformer | 32 s | 7.2 h | 6.6 ms | 853 K | 206 K |
| —— w/o $\psi$ / $\beta$ | Transformer | 12 s | 2.6 h | 5.9 ms | 820 K | 198 K |

Table 4: **Ablation studies on the validation sets of the JARVIS-DFT 3D 2021 dataset.**

| Settings | $\psi$ | # Blocks | Form. E. | Total E. | Bandgap (OPT) | Bandgap (MBJ) | E hull |
|---|---|---|---|---|---|---|---|
| Proposed | ✓ | 4 | **0.0301** | **0.0314** | **0.133** | **0.287** | **0.0487** |
| Simplified | | 4 | 0.0541 | 0.0546 | 0.140 | 0.308 | 0.0517 |

PotNet and Matformer, respectively. We believe that our neural potential summation induces a strong inductive bias and leads to a more compact model. Although our total training time is relatively long, our inference speed is the fastest. Note that PotNet reduces its training time by precomputing infinite potential summations for training data. For reference, we also show the efficiency of our model without $\psi/\beta$, indicating that the computations for $\beta$ largely dominate the entire training process.

## 5.4 ABLATION STUDIES

We compare the proposed model with its simplified version removing value position encoding $\psi$, as a variant close to Graphormer (Ying et al., 2021). Table 4 shows consistent performance improvements

by the inclusion of $\psi$, most notably on the formation and total energy predictions but less so on the others. In Appendix I, we discuss this behavior in terms of the definitions of these metrics, and also show further performance improvements by changing the number of self-attention blocks.

## 6 LIMITATIONS AND DISCUSSION

**Angular and directional information.** We currently adopt fully distance-based formulations for position econdings $\phi$ and $\psi$ to ensure SE(3) invariance. Although straightforward, such formulations limit the expressive power and the addition of angular/directional information is preferred (Pozdnyakov & Ceriotti, 2022). Some existing works on SE(3)-invariant GNNs explore this direction by using 3-body interactions (Park & Wolverton, 2020; Choudhary & DeCost, 2021; Chen & Ong, 2022) or plane-wave-based edge features (Cheng et al., 2021). Others propose SE(3)-equivariant encoding techniques for a similar purpose (Liao & Smidt, 2023; Duval et al., 2023). Extending Crystalformer to a 3-body, higher-order Transformer (Kim et al., 2021), incorporating plane wave features into $\psi$, or introducing SE(3)-equivariant transformations are possible future directions.

**Forms of interatomic potentials.** Explicitly exposing distance-decay functions in attention helps the physical interpretation of our model. Our current setting limits them to the Gaussian decay function, intending to approximate the sum of various potentials in the real world, such as the Coulomb potential ($1/r$) and the van der Waals potential ($1/r^6$). Despite the simplicity, the experiments show empirically good results, and we believe that the overall Transformer architecture (*i.e.*, MHA, FFN, and repeated attention blocks) helps to learn more complex potential forms than the Gaussian function. Still, we can explore other choices of functions by explicitly incorporating known potential forms into our model. This can be done by assigning different forms of $\exp(\phi)$ for different heads of MHA, which possibly leads to further performance gains or model efficiency. We leave it as our future work.

**Attention in Fourier space for long-range interactions.** Gaussian tail length $\sigma_i$ in Eq. (7) is upper-bounded by a certain constant, $\sigma_{\mathrm{ub}} \simeq 2\text{Å}$, to avoid excessively long-tailed functions, thus making $\alpha$ and $\beta$ computationally tractable. Despite the empirical success, such a bound possibly overlooks the presence of long-range interactions, such as the Coulomb potential. Here, our infinite-series formulation becomes advantageous for computing such long-range interactions, with the help of *reciprocal space*. Reciprocal space, or Fourier space, is analogous to the frequency domain of time-dependent functions, and appears in the 3D Fourier transform of spatial functions. When Eq. (5) is written as $\alpha_{ij} = \log f(\boldsymbol{p}_j - \boldsymbol{p}_i)$ with spatial function $f(\boldsymbol{r})$, $f$ is a periodic function that can be expressed in reciprocal space via Fourier series. In reciprocal space, $f$'s infinite series of Gaussian functions of distances in real space becomes an infinite series of Gaussian functions of spatial frequencies. (See Appendix J for detailed formulations.) These two expressions are complementary in that short-tail and long-tail potentials decay rapidly in real and reciprocal space, respectively. We use these two expressions for parallel heads of each MHA, by computing their $\alpha$ and $\beta$ differently in real or reciprocal space. To ensure the tractability in reciprocal space, $\sigma_i$ should be lower-bounded as $\sigma_i > \sigma_{\mathrm{lb}}$. By setting the bounds for the two spaces as $\sigma_{\mathrm{lb}} < \sigma_{\mathrm{ub}}$, this dual-space MHA can cover the entire range of interactions in theory. As a tentative experiment, we evaluate this dual-space variant of Crystalformer on the JARVIS-DFT dataset. Results in Table A6 in Appendix J.4 show that adding the reciprocal-space attention results in worse performance for formation and total energy, comparable for bandgap, and much better for E hull predictions. Adaptively switching each MHA head between the two attention forms depending on the task will lead to stable improvements.

## 7 CONCLUSIONS

We have presented Crystalformer as a Transformer encoder for crystal structures. It stands on fully connected attention between periodic points, namely infinitely connected attention, with physically-motivated distance-decay attention to ensure the tractability. The method has shown successful results in various property prediction tasks with high model efficiency. We have further discussed its extension to a reciprocal space representation for efficiently computing long-range interatomic interactions. We hope that this simple and physically-motivated Transformer framework provides a perspective in terms of both ML and materials science to promote further interdisciplinary research. As Transformer-based large language models are revolutionizing AI, how Crystalformer with large models can absorb knowledge from large-scale material datasets is also an ambitious open question.

AUTHOR CONTRIBUTIONS

T.T. conceived the core idea of the method, implemented it, conducted the experiments, and wrote the paper and rebuttal with other authors. R.I. suggested the Fourier space attention, provided knowledge of physics simulation and materials science, and helped with the paper writing and rebuttal. Y.S. provided a codebase for crystal encoders, provided materials science knowledge, and helped with the experiments, survey of crystal encoders, and paper writing. N.C. provided ML knowledge and helped with the experiments, survey of invariant encoders, and paper writing. K.S. provided materials science knowledge and helped with the paper writing. Y.U. co-led our materials-related collaborative projects, provided ML knowledge, and helped with the experiments and paper writing. K.O. co-led our materials-related collaborative projects, provided materials science knowledge, and helped with the paper writing.

ACKNOWLEDGMENTS

This work is partly supported by JST-Mirai Program, Grant Number JPMJMI19G1. Y.S. was supported by JST ACT-I grant number JPMJPR18UE during the early phase of this project. The computational resource of AI Bridging Cloud Infrastructure (ABCI) provided by the National Institute of Advanced Industrial Science and Technology (AIST) was partly used for numerical experiments.

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

## A  DERIVATION OF PSEUDO-FINITE PERIODIC ATTENTION

Starting from Eq. (3), we can show that

$$\boldsymbol{y}_i = \frac{1}{Z_i} \sum_{j=1}^{N} \sum_{\boldsymbol{n}} \exp\left(\boldsymbol{q}_i^T \boldsymbol{k}_{j(\boldsymbol{n})} / \sqrt{d_K} + \phi_{ij(\boldsymbol{n})}\right)\left(\boldsymbol{v}_{j(\boldsymbol{n})} + \boldsymbol{\psi}_{ij(\boldsymbol{n})}\right) \tag{A1}$$

$$= \frac{1}{Z_i} \sum_{j=1}^{N} \sum_{\boldsymbol{n}} \exp\left(\boldsymbol{q}_i^T \boldsymbol{k}_j / \sqrt{d_K} + \phi_{ij(\boldsymbol{n})}\right)\left(\boldsymbol{v}_j + \boldsymbol{\psi}_{ij(\boldsymbol{n})}\right) \tag{A2}$$

$$= \frac{1}{Z_i} \sum_{j=1}^{N} \exp\left(\boldsymbol{q}_i^T \boldsymbol{k}_j / \sqrt{d_K}\right) \sum_{\boldsymbol{n}} \exp(\phi_{ij(\boldsymbol{n})})(\boldsymbol{v}_j + \boldsymbol{\psi}_{ij(\boldsymbol{n})}) \tag{A3}$$

$$= \frac{1}{Z_i} \sum_{j=1}^{N} \exp\left(\boldsymbol{q}_i^T \boldsymbol{k}_j / \sqrt{d_K}\right) \exp(\alpha_{ij})(\boldsymbol{v}_j + \frac{\sum_{\boldsymbol{n}} \exp(\phi_{ij(\boldsymbol{n})})\boldsymbol{\psi}_{ij(\boldsymbol{n})}}{\exp(\alpha_{ij})}) \tag{A4}$$

$$= \frac{1}{Z_i} \sum_{j=1}^{N} \exp\left(\boldsymbol{q}_i^T \boldsymbol{k}_j / \sqrt{d_K} + \alpha_{ij}\right)(\boldsymbol{v}_j + \boldsymbol{\beta}_{ij}), \tag{A5}$$

which equals Eq. (4).

## B  BOUND FOR APPROXIMATION ERROR

The computations of $\alpha_{ij}$ and $\boldsymbol{\beta}_{ij}$ involve the following infinite summation:

$$Z_{ij} = \sum_{\boldsymbol{n}} \exp\left(-\frac{\|\boldsymbol{p}_j + L\boldsymbol{n} - \boldsymbol{p}_i\|^2}{2\sigma_i^2}\right) \tag{A6}$$

where $L = [\boldsymbol{\ell}_1, \boldsymbol{\ell}_2, \boldsymbol{\ell}_3]$. As explained in the implementation details (Sec. 3.1), we approximate $Z_{ij}$ by finite summation:

$$\tilde{Z}_{ij} = \sum_{n_1=-R_1}^{R_1} \sum_{n_2=-R_2}^{R_2} \sum_{n_3=-R_3}^{R_3} \exp\left(-\frac{\|\boldsymbol{p}_j + L\boldsymbol{n} - \boldsymbol{p}_i\|^2}{2\sigma_i^2}\right), \tag{A7}$$

where ranges $R_1, R_2, R_3 \in \mathbb{Z}_+$ are adaptively determined so that the summed points (*i.e.*, $\boldsymbol{p}_j + L\boldsymbol{n} - \boldsymbol{p}_i$) sufficiently cover a spherical radius of $c\sigma_i$ in 3D space ($c$ is set to 3.5) while ensuring $R_1, R_2, R_3 \geq 2$. This derives an upper bound for the residual error between $Z_{ij}$ and $\tilde{Z}_{ij}$ as

$$\epsilon_{ij} \triangleq Z_{ij} - \tilde{Z}_{ij} \leq \sum_{\boldsymbol{r} \in \Lambda_{ij}, \|\boldsymbol{r}\| \geq R} \exp\left(-\|\boldsymbol{r}\|^2\right), \tag{A8}$$

where $\Lambda_{ij} = \{\frac{1}{\sqrt{2}\sigma_i}(L\boldsymbol{n} + \boldsymbol{p}_j - \boldsymbol{p}_i) | \boldsymbol{n} \in \mathbb{Z}^3\}$, $R = \max(c/\sqrt{2}, R')$, $R' = \min\{\frac{1}{\sqrt{2}\sigma_i}\|\boldsymbol{p}_j + L\boldsymbol{n} - \boldsymbol{p}_i\| \,|\, \boldsymbol{n} \in \mathbb{Z}^3, \max(|n_1|, |n_2|, |n_3|) > 2\}$.

Here, Deconinck et al. (2004) prove the following inequality:

$$\sum_{\boldsymbol{r} \in \Lambda, \|\boldsymbol{r}\| \geq R} \|\boldsymbol{r}\|^p \exp\left(-\|\boldsymbol{r}\|^2\right) \leq \frac{d+p}{2}\left(\frac{2}{\rho}\right)^d \Gamma\left(\frac{d}{2}, \left(R - \frac{\rho}{2}\right)^2\right), \tag{A9}$$

where $\Lambda = \{X\boldsymbol{n} + \boldsymbol{x} | \boldsymbol{n} \in \mathbb{Z}^d\}$, $X \in \mathbb{R}^{d \times d}$ is a full-rank matrix, $\boldsymbol{x} \in \mathbb{R}^d$, $\rho = \min\{\|X\boldsymbol{n}\| \,|\, \boldsymbol{n} \in \mathbb{Z}^d, \boldsymbol{n} \neq \boldsymbol{0}\}$, and $\Gamma(z, x) = \int_x^{\infty} t^{z-1} e^{-t} dt$, the incomplete Gamma function.

Substituting $p = 0$, $d = 3$, and $X = \frac{1}{\sqrt{2}\sigma_i} L$ into this inequality derives a bound for our residual error:

$$\epsilon_{ij} \leq \frac{3}{2}\left(\frac{2}{\rho}\right)^3 \Gamma\left(\frac{3}{2}, \left(R - \frac{\rho}{2}\right)^2\right), \tag{A10}$$

where $\rho = \min\{\frac{1}{\sqrt{2}\sigma_i}\|L\boldsymbol{n}\| \,|\, \boldsymbol{n} \in \mathbb{Z}^d, \boldsymbol{n} \neq \boldsymbol{0}\}$.

## C  DEFINITION OF $\sigma_i$.

To increase the expressive power of the model, we treat tail length $\sigma_i$ of the Gaussian potential in Eq. (7) as a variable parameterized via current state $\boldsymbol{q}_i$. Here, the exponent of the Gaussian function, $\phi_{ij(\boldsymbol{n})} = -\|\boldsymbol{p}_{j(\boldsymbol{n})} - \boldsymbol{p}_i\|^2/2\sigma_i^2$, is intended to represent the relative position encoding by Shaw et al. (2018), whose original form is $\phi_{ij} = \boldsymbol{q}_i^T \boldsymbol{a}_{ij}$ with embedding $\boldsymbol{a}_{ij}$ of relative position $j - i$. As such a linear form of $\boldsymbol{q}_i$ should ease the training of the network, we model $\sigma_i^{-2}$ instead of $\sigma_i$ as a function:

$$\sigma_i^{-2} = r_0^{-2}\rho((\boldsymbol{q}_i^T \boldsymbol{w} - m)/s) \tag{A11}$$

where $r_0$ is a hyperparameter determining a distance scale of spatial dependencies in Å, $\boldsymbol{w}$ is a trainable weight vector, $m$ and $s$ are constant parameters, and $\rho : \mathbb{R} \to \mathbb{R}^+$ is a function to ensure a positive value for $\sigma_i^{-2}$.

The adjustment of $r_0$ could be difficult, since it has to bridge between network's internal states (*i.e.*, $\boldsymbol{q}_i^T \boldsymbol{w}$) and physical distances (*i.e.*, $\sigma_i$) across different scales. To ease it, we design $\rho$ so that the initial values of $\rho$ and hence $\sigma_i$ are distributed around 1 and $r_0$, respectively. To this end, we initialize $m$ and $s$ as the mean and standard deviation of the values of $\boldsymbol{q}_i^T \boldsymbol{w}$ computed for the mini-batch samples at the initial training step. Then, for $\rho$ we introduce a *shifted exponential linear unit* (shifted ELU):

$$\rho(x; a, b) = (1 - b)\text{ELU}(ax/(1 - b)) + 1, \tag{A12}$$

where $\text{ELU}(x) = \exp(x) - 1$ if $x < 0$ and $\text{ELU}(x) = x$ otherwise. Assuming that $x$ initially follows $\mathcal{N}(0, 1)$, shifted ELU $\rho(x; a, b)$ is designed to be 1) $\rho(0) = 1$ so that $\sigma_i$ is initially distributed around $r_0$, 2) lower-bounded as $\rho(x) > b$ so that $\sigma_i < r_0/\sqrt{b}$ to avoid excessively long tails, and 3) linear response with derivative $a$ for $x \geq 0$ to increase the training stability. We empirically set the hyperparameters to $(r_0, a, b) = (1.4, 0.1, 0.5)$.

## D  NECESSITY OF VALUE POSITION ENCODING

Shaw et al. (2018) discuss that value position encoding is required depending on the application, and we show that crystal structure encoding is indeed such a case.

As mentioned in the main texts, Eq. (3) without value position encoding $\boldsymbol{\psi}_{ij(\boldsymbol{n})}$ cannot distinguish between crystal structures of the same single atom in differently sized unit cells. This can be confirmed by substituting $\mathcal{X} = \{\boldsymbol{x}_1\}$ or $N = 1$ into Eq. (4) as below.

$$\boldsymbol{y}_1 = \frac{1}{Z_1}\exp\left(\boldsymbol{q}_1^T \boldsymbol{k}_1/\sqrt{d_K} + \alpha_{11}\right)(\boldsymbol{v}_1 + \boldsymbol{\beta}_{11}) \tag{A13}$$

$$= \boldsymbol{v}_1 + \boldsymbol{\beta}_{11} \tag{A14}$$

$$= \boldsymbol{v}_1 + \frac{\sum_{\boldsymbol{n}}\exp(\phi_{11(\boldsymbol{n})})\boldsymbol{\psi}_{11(\boldsymbol{n})}}{\sum_{\boldsymbol{n}}\exp(\phi_{11(\boldsymbol{n})})} \tag{A15}$$

Note that the lattice information (*i.e.*, $\boldsymbol{\ell}_1, \boldsymbol{\ell}_2, \boldsymbol{\ell}_3$) is used only in key position encoding $\phi_{11(\boldsymbol{n})}$ and value position encoding $\boldsymbol{\psi}_{11(\boldsymbol{n})}$ for $\alpha_{11}$ and $\boldsymbol{\beta}_{11}$, and does not exist in $\boldsymbol{q}, \boldsymbol{k}, \boldsymbol{v}$ in the initial state. The above equations show that when $N = 1$, the weight of $(\boldsymbol{v}_1 + \boldsymbol{\beta}_{11})$ is canceled out by $Z_1$ and therefore the lattice information is only present in $\boldsymbol{\beta}_{11}$. Without value position encoding (*i.e.*, $\boldsymbol{\psi}_{11(\boldsymbol{n})} = \boldsymbol{0}$), the attention formula becomes $\boldsymbol{y}_1 = \boldsymbol{v}_1$, showing that result $\boldsymbol{y}_1$ does not reflect how the atom is repeated in 3D space by $\boldsymbol{\ell}_1, \boldsymbol{\ell}_2, \boldsymbol{\ell}_3$.

From a materials science perspective, crystal structures of single unit-cell atoms are called *monatomic structures* and represent an important class of existing materials. Well-known monatomic structures of the same atom include graphite and diamond. They have the single unit-cell atom of Carbon but have different lattice vectors and thus different crystal structures. These differences make graphite a soft substance used for pencil tops while making diamond the hardest substance in the world. The value position encoding is thus required to distinguish such important materials.

## E  DETAILED NETWORK ARCHITECTURE

Table A1 shows the detailed specifications of the proposed Crystalformer architecture. Note that changing the ReLU activation function to GELU in the FFNs of the self-attention blocks did not lead to meaningful performance improvements.

Table A1: **Detailed network architecture.**

| Blocks | Output dims | Specifications |
|---|---|---|
| Atom embedding | $(128, N)$ | $\{e_k^{\text{atom}}\|k = \text{AtomicNumber}(i), i = 1, 2, ..., N\}$ 
 where $e^{\text{atom}}$'s are initialized by $\mathcal{N}(0, 128^{-0.5})$ |
| Self-attention block $\times$ 4 | $(128, N)$ | Model dim $d$: 128 
 Number of heads: 8 
 Query/Key/Value dims: 16 
 FFN: Linear-ReLU-Linear (dims: $128 \to 512 \to 128$) 
 Activation func: ReLU 
 Normalization: No |
| Pooling | 128 | Global average pooling |
| Feed forward | 1 | Linear with output dim: 128 
 ReLU 
 Linear with output dim: 1 (target property dim.) |

## F   INVARIANCE PROPERTIES

### F.1   DEFINITIONS

Following notation by Yan et al. (2022), we represent a crystal structure with a tuple of three matrices, $(X, P, L)$, where $X = [\boldsymbol{x}_1, ..., \boldsymbol{x}_N] \in \mathbb{R}^{d \times N}$ is the states of $N$ unit-cell atoms, $P = [\boldsymbol{p}_1, ..., \boldsymbol{p}_N] \in \mathbb{R}^{3 \times N}$ is the 3D Cartesian coordinates of these atoms, and $L = [\boldsymbol{\ell}_1, \boldsymbol{\ell}_2, \boldsymbol{\ell}_3] \in \mathbb{R}^{3 \times 3}$ is the lattice vector matrix. The coordinates of the $N$ unit-cell points are defined to be within the unit cell region defined by $L$, that is, they have fractional coordinates of $L^{-1}P \in [0, 1)^{3 \times N}$. When the overall network architecture is seen as function $f(X, P, L) \to \mathcal{X}$, they satisfy the following invariance properties.

**Definition 1: Unit cell permutation invariance.**   *A function $f : (X, P, L) \to \mathcal{X}$ is unit cell permutation invariant such that for any permutation function $\sigma : \{1, 2, .., N\} \to \{1, 2, .., N\}$, we have $f(X, P, L) = f(\sigma(X), \sigma(P), L)$, where $\sigma(X) = [\boldsymbol{x}_{\sigma(1)}, ..., \boldsymbol{x}_{\sigma(N)}]$ and $\sigma(P) = [\boldsymbol{p}_{\sigma(1)}, ..., \boldsymbol{p}_{\sigma(N)}]$.*

**Definition 2: Unit cell E(3) invariance.**   *A function $f : (X, P, L) \to \mathcal{X}$ is unit cell E(3) invariant such that for all $R \in \mathbb{R}^{3 \times 3}$, $R^T = R^{-1}$, $\det(R) = \pm 1$, and $\boldsymbol{b} \in \mathbb{R}^3$, we have $f(X, P, L) = f(X, T\bar{P}, RL)$, where $T = [R, \boldsymbol{b}] \in \mathbb{R}^{3 \times 4}$ is a rigid transformation, $R$ is a rotation matrix, $\boldsymbol{b}$ is a translation vector, and $\bar{P} \in \mathbb{R}^{4 \times N}$ is the homogeneous coordinates of $P$ in 3D space. (Note that the definition gives unit cell SE(3) invariance when $R$ is restricted to rotation matrix, as $\det(R) = 1$.)*

Due to the arbitrary choice of extracting a unit cell structure from the infinitely expanding crystal structure in 3D space, there is an infinite number of invariants $(\hat{X}, \hat{P}, \hat{L})$ that represent the same crystal structure as $(X, P, L)$. Similarly to the notation by Yan et al. (2022), we define such a unit-cell slice of a given crystal structure as

$$(\hat{X}, \hat{P}, \hat{L}) = \Phi(X, P, L, \boldsymbol{k}, \boldsymbol{p}). \tag{A16}$$

The slice is parameterized via a periodic boundary scale, $\boldsymbol{k} \in \mathbb{Z}_+^3$, that defines a supercell of $L$ as

$$\hat{L} = [k_1 \boldsymbol{\ell}_1, k_2 \boldsymbol{\ell}_2, k_3 \boldsymbol{\ell}_3], \tag{A17}$$

and a periodic boundary shift, $\boldsymbol{p} \in \mathbb{R}^3$, that provides a corner point of the periodic boundaries. The unit-cell atoms in the slice are then provided as a collection of the atoms contained in new unit cell $\hat{L}$ with the scaled and shifted periodic boundaries as

$$\hat{P} = \{\boldsymbol{p}_i + L\boldsymbol{n} - \boldsymbol{p}\|\hat{L}^{-1}(\boldsymbol{p}_i + L\boldsymbol{n} - \boldsymbol{p}) \in [0, 1)^3, i \in \{1, ..., N\}, \boldsymbol{n} \in \mathbb{Z}^3\}, \tag{A18}$$

$$\hat{X} = \{\boldsymbol{x}_i|\hat{L}^{-1}(\boldsymbol{p}_i + L\boldsymbol{n} - \boldsymbol{p}) \in [0, 1)^3, i \in \{1, ..., N\}, \boldsymbol{n} \in \mathbb{Z}^3\}. \tag{A19}$$

The periodic invariance described by Yan et al. (2022) is then summarized as follow.

**Definition 3: Periodic invariance.** *A unit cell E(3) invariant function $f : (X, P, L) \to \mathcal{X}$ is periodic invariant if $f(X, P, L) = f(\Phi(X, P, L, \boldsymbol{k}, \boldsymbol{p}))$ holds for all $\boldsymbol{k} \in \mathbb{Z}_+^3$ and $\boldsymbol{p} \in \mathbb{R}^3$.*

### F.2 PROOFS OF INVARIANCE

**Proof of unit cell permutation invariance.** Let us represent overall network transformation $f$ as

$$f(X, P, L) = o(h(g(X, P, L))), \tag{A20}$$

where $X' = g(X, P, L)$ is the transformation by stacked self-attention blocks, $\boldsymbol{z} = h(X')$ is a global pooling operation, and $o(\boldsymbol{z})$ is the final output network given global feature vector $\boldsymbol{z}$. We can easily show that $g$ is *permutation equivariant* such that $\sigma(X') = g(\sigma(X), \sigma(P), L)$ holds for any permutation $\sigma$. (This is obvious from that fact that substituting $i \leftarrow \sigma(i)$ and $j(\boldsymbol{n}) \leftarrow \sigma(j)(\boldsymbol{n})$ into the right-hand side of Eq. (3) yields $y_{\sigma(i)}$ and thus the attention operation by Eq. (3) is permutation equivariant.) We can then prove that $f(\sigma(X), \sigma(P), L) = o(h(\sigma(X'))) = o(h(X)) = f(X, P, L)$ holds if $h$ is a permutation invariant pooling operation such as the global mean or max pooling.

**Proof of unit cell E(3) invariance.** In $f$, the spatial information (*i.e.*, unit-cell atom positions $P$ and lattice vectors $L$) is only used by the position encodings, $\phi_{ij(\boldsymbol{n})}$ and $\psi_{ij(\boldsymbol{n})}$, in Eq. (3). Currently, they are defined to simply encode relative distances as

$$\phi_{ij(\boldsymbol{n})} = \phi(\|\boldsymbol{p}_i - \boldsymbol{p}_{j(\boldsymbol{n})}\|), \tag{A21}$$

$$\boldsymbol{\psi}_{ij(\boldsymbol{n})} = \psi(\|\boldsymbol{p}_i - \boldsymbol{p}_{j(\boldsymbol{n})}\|), \tag{A22}$$

thus invariant to rigid transformation $(P', L') \leftarrow (T\bar{P}, RL)$ as

$$\|\boldsymbol{p}'_i - \boldsymbol{p}'_{j(\boldsymbol{n})}\| = \|\boldsymbol{p}'_i - (\boldsymbol{p}'_j + L'\boldsymbol{n})\| \tag{A23}$$

$$= \|(R\boldsymbol{p}_i + \boldsymbol{b}) - (R\boldsymbol{p}_j + \boldsymbol{b} + RL\boldsymbol{n})\| \tag{A24}$$

$$= \|R(\boldsymbol{p}_i - \boldsymbol{p}_{j(\boldsymbol{n})})\| \tag{A25}$$

$$= \|\boldsymbol{p}_i - \boldsymbol{p}_{j(\boldsymbol{n})}\|. \tag{A26}$$

Note that our framework allows the use of directional information by using relative position ($\boldsymbol{p}_i - \boldsymbol{p}_{j(\boldsymbol{n})}$) instead of relative distance $\|\boldsymbol{p}_i - \boldsymbol{p}_{j(\boldsymbol{n})}\|$, provided that $\phi(\boldsymbol{r})$ and $\psi(\boldsymbol{r})$ are rotation invariant.

**Proof of periodic invariance.** It is sufficient to show that Eq. (3) is invariant for minimum repeatable unit-cell representation $(X, P, L)$ and its invariant representation $(\hat{X}, \hat{P}, \hat{L}) = \Phi(X, P, L, \boldsymbol{k}, \boldsymbol{p})$. Periodic invariance of Eq. (3) is intuitively understandable, because Eq. (3) recovers the original, infinitely expanding crystal structure from its unit cell representation and attends to all the atoms in it with relative position encoding. A formal proof is provided below.

Comparing $(X, P, L)$ and $(\hat{X}, \hat{P}, \hat{L})$, each unit-cell point $i \in \{1, ..., N\}$ in $(X, P, L)$ is populated to $k_1 k_2 k_3$ unit-cell points in $(\hat{X}, \hat{P}, \hat{L})$, which we index by $i[\boldsymbol{m}]$ where $\boldsymbol{m} = (m_1, m_2, m_3)^T \in \mathbb{Z}^3$ and $0 \le m_1 \le k_1 - 1, 0 \le m_2 \le k_2 - 1, 0 \le m_3 \le k_3 - 1$ (or simply $\boldsymbol{0} \le \boldsymbol{m} \le \boldsymbol{k} - \boldsymbol{1}$). These populated points $i[\boldsymbol{m}]$ in $(\hat{X}, \hat{P}, \hat{L})$ have the same state:

$$\hat{\boldsymbol{x}}_{i[\boldsymbol{m}]} = \boldsymbol{x}_i, \tag{A27}$$

and periodically shifted positions:

$$\hat{\boldsymbol{p}}_{i[\boldsymbol{m}]} = \boldsymbol{p}_i + L(\boldsymbol{m} + \boldsymbol{c}_i) - \boldsymbol{p}, \tag{A28}$$

where $\boldsymbol{c}_i \in \mathbb{Z}^3$ is a constant chosen appropriately for each $i$ so that $\hat{\boldsymbol{p}}_{i[\boldsymbol{m}]}$ for all $\boldsymbol{m}: \boldsymbol{0} \le \boldsymbol{m} \le \boldsymbol{k} - \boldsymbol{1}$ reside within the unit cell region of $\hat{L}$ (*i.e.*, $\hat{L}^{-1}\hat{\boldsymbol{p}}_{i[\boldsymbol{m}]} \in [0, 1)^3$).

By simply applying Eq. (3) for $(\hat{X}, \hat{P}, \hat{L})$, we obtain

$$\hat{\boldsymbol{y}}_i = \frac{1}{\hat{Z}_i} \sum_{j=1}^{k_1 k_2 k_3 N} \sum_{\boldsymbol{n}} \exp\left(\hat{\boldsymbol{q}}_i^T \hat{\boldsymbol{k}}_j / \sqrt{d_K} + \hat{\phi}_{ij(\boldsymbol{n})}\right)\left(\hat{\boldsymbol{v}}_j + \hat{\boldsymbol{\psi}}_{ij(\boldsymbol{n})}\right), \tag{A29}$$

where $\hat{q}$, $\hat{k}$, and $\hat{v}$ are key, query, and value of $\hat{x}$ in $\hat{X}$. Scalar $\hat{\phi}_{ij(n)}$ and vector $\hat{\psi}_{ij(n)}$ encode the interatomic spatial relation as $\phi(\hat{p}_{j(n)} - \hat{p}_i)$ and $\psi(\hat{p}_{j(n)} - \hat{p}_i)$, respectively, where $\hat{p}_{j(n)} = \hat{p}_j + \hat{L}n$. We rewrite Eq. (A29) by rewriting populated point $i$ as $i[m]$ and $j$ as $j[m']$, obtaining

$$\hat{y}_{i[m]} = \frac{1}{\hat{Z}_{i[m]}} \sum_{j=1}^{N} \sum_{m'} \sum_{n} \exp\left(\hat{q}_{i[m]}^T \hat{k}_{j[m']} / \sqrt{d_K} + \hat{\phi}_{i[m]j[m'](n)}\right)\left(\hat{v}_{j[m']} + \hat{\psi}_{i[m]j[m'](n)}\right),$$
(A30)

where $\sum_{m'}$ is short for $\sum_{m_1'=0}^{k_1-1} \sum_{m_2'=0}^{k_2-1} \sum_{m_3'=0}^{k_3-1}$ with $m' = (m_1', m_2', m_3')^T$. Since $\hat{x}_{i[m]}$ in $\hat{X}$ equals $x_i$ in $X$, we substitute $\hat{q}_{i[m]} = q_i$, $\hat{k}_{j[m']} = k_j$, and $\hat{v}_{j[m']} = v_j$ to obtain

$$\hat{y}_{i[m]} = \frac{1}{\hat{Z}_{i[m]}} \sum_{j=1}^{N} \sum_{m'} \sum_{n} \exp\left(q_i^T k_j / \sqrt{d_K} + \hat{\phi}_{i[m]j[m'](n)}\right)\left(v_j + \hat{\psi}_{i[m]j[m'](n)}\right).$$
(A31)

Similarly to Eq. (4), this infinitely connected attention can be written in the following pseudo-finite periodic attention form:

$$\hat{y}_{i[m]} = \frac{1}{\hat{Z}_{i[m]}} \sum_{j=1}^{N} \exp\left(q_i^T k_j / \sqrt{d_K} + \hat{\alpha}_{i[m]j}\right)(v_j + \hat{\beta}_{i[m]j}),$$
(A32)

where

$$\hat{\alpha}_{i[m]j} = \log \sum_{n} \sum_{m'} \exp\left(\hat{\phi}_{i[m]j[m'](n)}\right),$$
(A33)

$$\hat{\beta}_{i[m]j} = \frac{1}{\hat{Z}_{i[m]j}} \sum_{n} \sum_{m'} \exp(\hat{\phi}_{i[m]j[m'](n)})\hat{\psi}_{i[m]j[m'](n)},$$
(A34)

and $\hat{Z}_{i[m]j} = \exp(\hat{\alpha}_{i[m]j})$. Here, $\hat{\phi}_{i[m]j[m'](n)}$ encodes the interatomic spatial relation:

$$\hat{p}_{j[m'](n)} - \hat{p}_{i[m]} = (\hat{p}_{j[m']} + \hat{L}n) - \hat{p}_{i[m]}$$
(A35)

$$= (p_j + Lm' + Lc_j - p + \hat{L}n) - (p_i + Lm + Lc_i - p)$$
(A36)

$$= p_j + (\hat{L}n + Lm') + L(c_j - m - c_i) - p_i.$$
(A37)

Notice here that

$$\hat{L}n + Lm' = (n_1 k_1 + m_1')\ell_1 + (n_2 k_2 + m_2')\ell_2 + (n_3 k_3 + m_3')\ell_3$$
(A38)

$$= Ln',$$
(A39)

where $n' = (n_1 k_1 + m_1', n_2 k_2 + m_2', n_3 k_3 + m_3')^T \in \mathbb{Z}^3$. Thus, we obtain

$$\hat{p}_{j[m'](n)} - \hat{p}_{i[m]} = p_j + L(n' + c) - p_i,$$
(A40)

where $c = (c_j - m - c_i) \in \mathbb{Z}^3$. Because $\sum_n \sum_{m'}$ for terms of $n'$ is equivalent to infinite series $\sum_{n'}$ over all $n' \in \mathbb{Z}^3$, we obtain the following equivalence:

$$\hat{\alpha}_{i[m]j} = \log \sum_{m'} \sum_{n} \exp\left(\phi(\hat{p}_{j[m'](n)} - \hat{p}_{i[m]})\right)$$
(A41)

$$= \log \sum_{n'} \exp\left(\phi(p_j + L(n' + c) - p_i)\right)$$
(A42)

$$= \log \sum_{n'} \exp\left(\phi_{ij(n'+c)}\right).$$
(A43)

Therefore, $\hat{\alpha}_{i[m]j}$ converges to the same value of $\alpha_{ij}$ in Eq. (5), if they converge. Likewise, $\hat{\beta}_{i[m]j}$ converges to the same value of $\beta_{ij}$ in Eq. (6). Provided that $\hat{\alpha}_{i[m]j} = \alpha_{ij}$ and $\hat{\beta}_{i[m]j} = \beta_{ij}$, the right-hand side of Eq. (A32) equals the right-hand side of Eq. (4), thus proving $\hat{y}_{i[m]} = y_i$.

At the overall architecture level, the above result indicates that applying the stacked self-attention blocks for an invariant representation, as $\hat{X}' = g(\Phi(X, P, L, k, p))$, gives $k_1 k_2 k_3$ duplicates of $X' = g(X, P, L)$. For these $\hat{X}'$ and $X'$, pooling operation $h$ such as global mean or max pooling holds the invariance: $h(\hat{X}') = h([X', ..., X']) = h(X')$. Thus, overall network transformation $f$ is periodic invariant.

## G   DIFFERENCE FROM POTNET

PotNet is a GNN-based crystal-structure encoder proposed by Lin et al. (2023). PotNet shares similarity with Crystalformer in that it uses infinite summations of interatomic potentials to model the periodicity of crystal structures. However, in addition to the obvious difference of being a GNN or a Transformer, *there is a more fundamental or conceptual difference in their representations of potential summations*. To see this, let us discuss how these methods perform state evolution.

**State evolution in PotNet.**   Using our notation, the PotNet's state evolution can be essentially expressed as

$$\boldsymbol{x}_i' \leftarrow g\left(\boldsymbol{x}_i, \sum_{j=1}^{N} f(\boldsymbol{x}_i, \boldsymbol{x}_j, \boldsymbol{e}_{ij})\right), \tag{A44}$$

where $f$ and $g$ are learnable GNN layers that perform message passing on all pairs of hidden state variables $(\boldsymbol{x}_i, \boldsymbol{x}_j)$ and edge features $\boldsymbol{e}_{ij}$ between them. Thus, PotNet presents itself in a standard GNN framework for fully connected graphs of nodes of unit-cell atoms. The most important part of PotNet is its edge feature $\boldsymbol{e}_{ij}$, which is an *embedding* of an infinite summation of scalar potentials. This summation is defined for each unit-cell atom pair $(i, j)$ as

$$S_{ij} = \sum_{\boldsymbol{n}} \Phi(\|\boldsymbol{p}_{j(\boldsymbol{n})} - \boldsymbol{p}\|) \tag{A45}$$

using a scalar potential function $\Phi(r) : \mathbb{R}_+ \rightarrow \mathbb{R}$. PotNet exploits three known forms of actual interatomic potentials, that is, Coulomb potential $1/r$, London dispersion potential $1/r^6$, and Pauli repulsion potential $e^{-\alpha r}$, via their summation:

$$\Phi(r) = \kappa_1/r + \kappa_2/r^6 + \kappa_3 e^{-\alpha r}, \tag{A46}$$

where $\kappa_1, \kappa_2, \kappa_3, \alpha$ are hyperparameters. Edge feature $\boldsymbol{e}_{ij}$ is then provided as its embedding expanded and encoded via radial basis functions (RBF) and multi-layer perceptron (MLP) as

$$\boldsymbol{e}_{ij} = \text{MLP}(\text{RBF}(S_{ij})). \tag{A47}$$

Here, we see that *PotNet exploits the infinite summations of known interatomic potentials to derive physics- and periodicity-informed edge features $\boldsymbol{e}_{ij}$, to be processed in a standard GNN framework.*

**State evolution in Crystalformer.**   The core part of our state evolution is the attention operation provided in Eq. (3) and written again below:

$$\boldsymbol{y}_i = \frac{1}{Z_i} \sum_{j=1}^{N} \sum_{\boldsymbol{n}} \exp(\boldsymbol{q}_i^T \boldsymbol{k}_j / \sqrt{d_K} + \phi_{ij(\boldsymbol{n})})(\boldsymbol{v}_j + \boldsymbol{\psi}_{ij(\boldsymbol{n})}) \tag{A48}$$

As described in Sec. 3.1, *this attention operation, as a whole, is interpreted as an infinite summation of interatomic potentials defined and performed in an abstract feature space.*

One may find some similarities of our $\alpha_{ij}$ and $\boldsymbol{\beta}_{ij}$ with $S_{ij}$ and $\boldsymbol{e}_{ij}$ of PotNet, because they enable a standard fully-connected Transformer or a fully-connected GNN to incorporate the crystal periodicity. However, our $\alpha_{ij}$ and $\boldsymbol{\beta}_{ij}$ are rather bi-products of the explicit potential summation in feature space performed as the infinitely connected attention, whereas $S_{ij}$ is a potential summation calculated in potential space before the GNN message passing (typically as preprocessing) and $\boldsymbol{e}_{ij}$ is an abstract embedding of $S_{ij}$.

**Summary.**

- PotNet proposes a new type of physics-informed edge feature by using the values of infinite summations of interatomic potentials. This edge feature implicitly embeds the information of crystal periodicity and can thus allow a standard fully-connected GNN to be informed of crystal periodicity.

- Crystalformer deeply fuses the calculations of infinite potential summations with neural networks, by performing the infinitely connected attention as an infinite potential summation expressed in abstract feature space. This attention operation can be performed in a standard fully-connected Transformer by using proposed position encoding $\alpha_{ij}$ and $\boldsymbol{\beta}_{ij}$.

## H    Hyperparameters for the JARVIS-DFT dataset

Compared to the Materials Project dataset, the JARVIS-DFT dataset is relatively small in both its training set size and average system size (*i.e.*, average number of atoms in unit cells, which is approximately 10 atoms in the JARVIS-DFT while 30 in the Materials Project). To account for these differences, we increase the batch size and number of total epochs as shown in Table A2.

Table A2: **Hyperparameters for the JARVIS-DFT dataset.**

|                   | Form. energy | Total energy | Bandgap (OPT) | Bandgap (MBJ) | E hull |
|-------------------|--------------|--------------|---------------|---------------|--------|
| Training set size | 44578        | 44578        | 44578         | 14537         | 44296  |
| Batch size        | 256          | 256          | 256           | 256           | 256    |
| Total epochs      | 800          | 800          | 800           | 1600          | 800    |

## I    Detailed ablation studies

### I.1    Physical interpretations of results in Table 4

Table 4 indicates the inclusion of the value position encoding yields greater improvements for the formation energy and total energy while relatively limited improvements for the bandgap and energy above hull (E hull). This result can be understood from the fact that *the formation energy and total energy are absolute energy metrics, whereas the bandgap and energy above hull are relative energy metrics that evaluate gaps between two energy levels, as* $\Delta E = E_1 - E_0$.

The accurate prediction of such energy gaps is generally difficult, because it needs accurate predictions of both $E_1$ and $E_0$. Furthermore, even when an algorithmic improvement in the prediction method leads to improvements in absolute energy prediction (e.g., formation energy prediction), it does not necessarily lead to improvements in energy gap prediction. Specifically, assume that an improved algorithm predicts energies $E_1' = E_1 + \Delta_1$ and $E_0' = E_0 + \Delta_0$ as better predictions than $E_1$ and $E_0$, respectively. These improvements can directly lead to lower prediction errors of absolute energy metrics. However, when improvements $\Delta_1$ and $\Delta_0$ are made by the same mechanism, it may be possible that $\Delta_1 \simeq \Delta_0$ so that they are canceled out in energy gaps $\Delta E' = E_1' - E_0'$.

We conjecture that the limited improvements in the bandgap and energy above hull prediction are due to these difficulties of energy gap prediction.

### I.2    Changing the number of self-attention blocks

We further study the effects of the number of self-attention blocks. We here utilize the simplified variant because of its relatively good performance and high efficiency (Table 3). The results in Table A3 suggest that the performance moderately mounts on a plateau with four or more blocks.

Table A3: **Ablation studies by changing the number of self-attention blocks.**

| Settings   | # Blocks | Form. E. | Total E. | Bandgap (OPT) | Bandgap (MBJ) | E hull  |
|------------|----------|----------|----------|---------------|---------------|---------|
| Simplified | 1        | 0.0664   | 0.0669   | 0.171         | 0.368         | 0.1313  |
|            | 2        | 0.0627   | 0.0609   | 0.153         | 0.329         | 0.0965  |
|            | 3        | 0.0558   | 0.0569   | 0.145         | 0.320         | 0.0676  |
|            | 4        | 0.0541   | 0.0546   | 0.140         | 0.308         | 0.0517  |
|            | 5        | 0.0525   | 0.0543   | 0.143         | 0.309         | 0.0484  |
|            | 6        | 0.0503   | 0.0533   | 0.140         | 0.323         | 0.0430  |
|            | 7        | **0.0498** | **0.0515** | **0.131**   | **0.298**     | **0.0407** |

Given the good performance of the seven-block setting, we test a seven-block variant of the proposed full model on the Materials Project and JARVIS-DFT datasets. The results in Tables A4 and A5 show that the larger model achieves slightly better accuracies, especially when the training set is large enough. Investigating how the performance changes with larger-scale datasets (Kirklin et al., 2015; Draxl & Scheffler, 2019) is left as a future topic.

Table A4: **Performance of the larger model on the Materials Project (MEGNet) dataset.**

|  | Formation energy 60000 / 5000 / 4239 | Bandgap 60000 / 5000 / 4239 | Bulk modulus 4664 / 393 / 393 | Shear modulus 4664 / 392 / 393 |
|---|---|---|---|---|
| 4 blocks (default) | 0.0186 | 0.198 | 0.0377 | **0.0689** |
| 7 blocks | **0.0185** | **0.184** | **0.0372** | 0.0717 |

Table A5: **Performance of the larger model on the JARVIS-DFT 3D 2021 dataset.**

|  | Form. energy 44578 / 5572 / 5572 | Total energy 44578 / 5572 / 5572 | Bandgap (OPT) 44578 / 5572 / 5572 | Bandgap (MBJ) 14537 / 1817 / 1817 | E hull 44296 / 5537 / 5537 |
|---|---|---|---|---|---|
| 4 blocks (default) | 0.0306 | 0.0320 | 0.128 | **0.274** | 0.0463 |
| 7 blocks | **0.0297** | **0.0312** | **0.127** | 0.275 | **0.0461** |

## J RECIPROCAL SPACE ATTENTION FOR LONG-RANGE INTERACTIONS

### J.1 BACKGROUND

Before presenting our attention formulation in reciprocal space, we briefly introduce its background from physical simulations. In physical simulations, the difficulty of computing long-range interactions in periodic systems is well known. Given potential function $\Phi(r)$ and physical quantities $\{v_1, ..., v_N\}$ (*e.g.*, electric charges) associated with unit-cell atoms, these simulations typically involve the summation of interatomic interactions:

$$V_i = \sum_{j(\boldsymbol{n}) \neq i} \Phi(\|\boldsymbol{p}_{j(\boldsymbol{n})} - \boldsymbol{p}_i\|) v_j, \tag{A49}$$

where $\sum_{j(\boldsymbol{n}) \neq i}$ sums for all the atoms $j(\boldsymbol{n})$ in the structure, as $\sum_{j=1}^{N} \sum_{\boldsymbol{n}}$, except for $j(\boldsymbol{n}) = i$. Its direct computation expanding $\boldsymbol{n}$ causes inefficiency for long-range $\Phi(r)$ due to slow convergence. Thus, DFT calculations often employ an efficient calculation method called Ewald summation.

In Ewald summation, potential $\Phi(r)$ is represented as the sum of short-range and long-range terms as

$$\Phi(r) = \Phi_{\text{short}}(r) + \Phi_{\text{long}}(r). \tag{A50}$$

In this manner, the summation of interactions is divided into two parts as

$$V_i = \sum_{j(\boldsymbol{n}) \neq i} \Phi_{\text{short}}(\|\boldsymbol{p}_{j(\boldsymbol{n})} - \boldsymbol{p}_i\|) v_j + \sum_{j(\boldsymbol{n}) \neq i} \Phi_{\text{long}}(\|\boldsymbol{p}_{j(\boldsymbol{n})} - \boldsymbol{p}_i\|) v_j. \tag{A51}$$

The first part is the sum of fast-decay potentials in distance, which can be efficiently computed directly in real space. On the other hand, long-range potentials have concentrated low-frequency components and thus have fast decay in frequency. Therefore, the second part can be efficiently computed in reciprocal space via its Fourier series expansion.

### J.2 RECIPROCAL SPACE ATTENTION

Kosmala et al. (2023) recently proposed Ewald message passing in the GNN framework, which performs graph convolutions in both real and reciprocal space. We also import the idea of Ewald summation into our Transformer framework.

To obtain reciprocal-space representations of $\alpha_{ij}$ and $\boldsymbol{\beta}_{ij}$ in Eqs. (5) and (6), it is sufficient to derive for infinite series $\sum_{\boldsymbol{n}} \exp(\phi_{ij(\boldsymbol{n})}) \boldsymbol{\psi}_{ij(\boldsymbol{n})}$ in the right-hand side of Eq. (6). We rewrite it as $f(\boldsymbol{p}_j - \boldsymbol{p}_i)$ using the following 3D spatial function:

$$f(\boldsymbol{r}) = \sum_{\boldsymbol{n}} \exp\left(\phi_i(\boldsymbol{r} + n_1 \boldsymbol{l}_1 + n_2 \boldsymbol{l}_2 + n_3 \boldsymbol{l}_3)\right) \boldsymbol{\psi}_i(\boldsymbol{r} + n_1 \boldsymbol{l}_1 + n_2 \boldsymbol{l}_2 + n_3 \boldsymbol{l}_3). \tag{A52}$$

With this spatial function, $\alpha_{ij}$ and $\boldsymbol{\beta}_{ij}$ are expressed as

$$\alpha_{ij} = \log f(\boldsymbol{p}_j - \boldsymbol{p}_i | \boldsymbol{\psi}_i \leftarrow 1), \tag{A53}$$

$$\boldsymbol{\beta}_{ij} = \frac{1}{Z_{ij}} f(\boldsymbol{p}_j - \boldsymbol{p}_i), \tag{A54}$$

where $f(\cdot|\boldsymbol{\psi}_i \leftarrow 1)$ denotes substitution $\boldsymbol{\psi}_i \leftarrow 1$ into Eq. (A52).

Since $f$ is a periodic function, it can be alternatively expressed via Fourier series expansion with $f$'s Fourier coefficient $F(\boldsymbol{\omega_m})$ as

$$f(\boldsymbol{r}) = \sum_{\boldsymbol{m} \in \mathbb{Z}^3} F(\boldsymbol{\omega_m}) \exp\left(i\boldsymbol{\omega_m} \cdot \boldsymbol{r}\right), \tag{A55}$$

where $\boldsymbol{i}$ is the imaginary unit and

$$\boldsymbol{\omega_m} = m_1\bar{\boldsymbol{l}}_1 + m_2\bar{\boldsymbol{l}}_2 + m_3\bar{\boldsymbol{l}}_3 \tag{A56}$$

is a 3D vector analogous to the angular frequency of the Fourier transform. $\boldsymbol{\omega_m}$ is defined with three integers $\boldsymbol{m} = (m_1, m_2, m_3) \in \mathbb{Z}^3$ with reciprocal lattice vectors:

$$(\bar{\boldsymbol{l}}_1, \bar{\boldsymbol{l}}_2, \bar{\boldsymbol{l}}_3) = (\frac{2\pi}{V}\boldsymbol{l}_2 \times \boldsymbol{l}_3, \frac{2\pi}{V}\boldsymbol{l}_3 \times \boldsymbol{l}_1, \frac{2\pi}{V}\boldsymbol{l}_1 \times \boldsymbol{l}_2) \tag{A57}$$

where $V = (\boldsymbol{l}_1 \times \boldsymbol{l}_2) \cdot \boldsymbol{l}_3$ is the cell volume. Fourier coefficient $F(\boldsymbol{\omega_m})$ is obtained via 3D integration:

$$F(\boldsymbol{\omega_m}) = \frac{1}{V} \iiint_{\mathbb{R}^3_{\text{cell}}} f(\boldsymbol{r}) \exp\left(-i\boldsymbol{\omega_m} \cdot \boldsymbol{r}\right) dx dy dz \tag{A58}$$

$$= \frac{1}{V} \iiint_{\mathbb{R}^3} \exp\left(\phi_i(\boldsymbol{r})\right) \boldsymbol{\psi}_i(\boldsymbol{r}) \exp\left(-i\boldsymbol{\omega_m} \cdot \boldsymbol{r}\right) dx dy dz, \tag{A59}$$

where $\mathbb{R}^3_{\text{cell}} \subset \mathbb{R}^3$ is the 3D domain of the unit cell. Intuitively, $F(\boldsymbol{\omega_m})$ with small or large $\|\boldsymbol{m}\|$ represents a low or high frequency component of $f(\boldsymbol{r})$.

Computing $\alpha_{ij}$ and $\boldsymbol{\beta}_{ij}$ with Eqs. (A53) and (A54) in reciprocal space using Fourier series in Eq. (A55) is effective when potential function $\exp(\phi_{ij(\boldsymbol{n})})$ is long-tailed, because $F(\boldsymbol{\omega_m})$ in such a case decays rapidly as $\|\boldsymbol{m}\|$ increases.

J.3    GAUSSIAN DISTANCE DECAY ATTENTION IN RECIPROCAL SPACE.

The reciprocal-space representation of $\alpha_{ij}$ in the case of the Gaussian distance decay function in Eq. (7) is expressed as follows.

$$\alpha_{ij} = \log\left[\frac{(2\pi\bar{\sigma}_i^2)^{3/2}}{V} \sum_{\boldsymbol{m}} \exp\left(-\frac{\bar{\sigma}_i^2\|\boldsymbol{\omega_m}\|^2}{2}\right) \cos\left(\boldsymbol{\omega_m} \cdot (\boldsymbol{p}_j - \boldsymbol{p}_i)\right)\right] \tag{A60}$$

Eq. (A60) is essentially an infinite series of Gaussian functions of frequencies $\|\boldsymbol{\omega_m}\|$, which rapidly decay as frequencies increase. On the other hand, we omit $\boldsymbol{\beta}_{ij}$ in the reciprocal-space attention by assuming $\boldsymbol{\psi}_{ij(\boldsymbol{n})} = \boldsymbol{\beta}_{ij} = \boldsymbol{0}$, because its analytic solution does not exist for our current $\boldsymbol{\psi}_{ij(\boldsymbol{n})}$ provided in Eq. (9). Exploring an effective form of $\boldsymbol{\psi}_{ij(\boldsymbol{n})}$ for the reciprocal-space attention is left as future work.

When $\bar{\sigma}_i^2 = \sigma_i^2$, the two expressions of $\alpha_{ij}$ with Eqs. (5) and (A60) become theoretically equivalent. Similarly to Ewald summation, however, we want to use the real-space attention for short-range interactions and the reciprocal-space attention for long-range interactions. To this end, we treat $\sigma_i$ and $\bar{\sigma}_i$ as independent variables by following a parameterization method in Appendix C. Specifically, we parameterize $\bar{\sigma}_i^2$ as

$$\bar{\sigma}_i^2 = \bar{r}_0^2 \rho((\boldsymbol{q}_i^T \bar{\boldsymbol{w}} - m)/s; \bar{a}, \bar{b}). \tag{A61}$$

In this way, $\bar{\sigma}_i$ is lower bounded as $\bar{\sigma}_i > \sigma_{\text{lb}}$ with $\sigma_{\text{lb}} = \bar{r}_0\sqrt{\bar{b}}$. We use $(\bar{r}_0, \bar{a}, \bar{b}) = (2.2, 0.1, 0.5)$ for $\bar{\sigma}_i$. Now $\sigma_i$ and $\bar{\sigma}_i$ have an upper and lower bound, respectively, as $\sigma_i < \sigma_{\text{ub}} \simeq 1.98\text{Å}$ and $\bar{\sigma}_i > \sigma_{\text{lb}} \simeq 1.56\text{Å}$. Therefore, *the combined use of the real-space and reciprocal-space attention can cover the entire range of interactions in principle.* We compute Eq. (A60) with a fixed range of $\|\boldsymbol{m}\|_\infty \leq 2$. Note that because of the symmetry, we can reduce the range of $m_1$ as $0 \leq m_1 \leq 2$ by doubling the summed values for $m_1 > 0$.

Table A6: **Effects of attention in the Fourier space on the JARVIS-DFT dataset.**

| Method | Form. energy | Total energy | Bandgap (OPT) | Bandgap (MBJ) | E hull |
|---|---|---|---|---|---|
| Crystalformer (real space) | **0.0306** | **0.0320** | 0.128 | **0.274** | 0.0463 |
| Crystalformer (dual space) | 0.0343 | 0.0353 | **0.126** | 0.283 | **0.0284** |

## J.4 RESULTS USING THE DUAL SPACE ATTENTION

As discussed in Sec. 6, we evaluate a variant of our method by changing half the heads of each MHA to the reciprocal attention provided in Appendix J.3. Table A6 shows the results with this dual-space attention.

We observe that the dual-space variant yields a significant improvement in the energy above hull prediction. Although its exact mechanism is unknown, we provide hypothetical interpretations below, as we do for results in Table 4 in Appendix I. In short, we conjecture that it is related to the fact that *the energy above hull is the only metric in Table A6 that involves the energy evaluation of different crystal structures than the targeted one*.

A more in-depth discussion needs to know the definition of the energy above hull. Let's again express it as $\Delta E = E_1 - E_0$, where $E_1$ corresponds to the formation energy of the target structure and $E_0$ is the so-called convex hull provided by stable phases of the target structure. As we know, substances around us change their phases between solid, liquid, and gas depending on the temperature and pressure. Likewise, a crystal structure has multiple stable phases with different structures within the solid state. For example, a structure with a chemical formula of $AB$ has phases that have different structures with chemical formulas of $A_x B_{1-x}$, $x \in [0, 1]$ (i.e., structures with the same element composition but different ratios of elements). When we plot $y = \text{FormationEnergy}(A_x B_{1-x})$ for all possible phase structures of $AB$, we can draw a convex hull of the plots, as in Fig. A1, which will have some corner points that represent stable phases of $AB$ (*i.e.*, S1 to S4 in Fig. A1). The energy above hull is then defined as the vertical deviation between the formation energy of the target structure and the line of the convex hull, essentially representing the instability of the target structure.

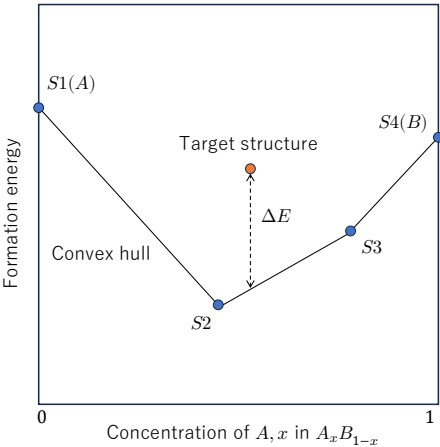

Figure A1: **Energy above hull.** Image courtesy of *Ma, J., Hegde, V., Munira, K., Xie, Y., Keshavarz, S., Mildebrath, D., Wolverton, C., Ghosh, A., & Butler, W. (2017). Computational investigation of half-Heusler compounds for spintronics applications. Phys. Rev. B, 95, 024411.*

When $E_1$ and $E_0$ involve the energy evaluation of different structures, it may be possible that the inclusion of long-range interactions has a positive impact on some of the stable phases of a given structure in $E_0$ while not so for the given structure itself in $E_1$. This asymmetric relation between $E_1$ and $E_0$ may mitigate the aforementioned cancellation effect and may lead to improved prediction of $E_0$ and $\Delta E$.

