# OpenReview forum: "Crystalformer: Infinitely Connected Attention for Periodic Structure Encoding"
_ICLR.cc/2024/Conference — ICLR 2024 poster_

### Official Review · Reviewer_8AQw · 2023-10-27

**Soundness:** 2 fair
**Presentation:** 3 good
**Contribution:** 2 fair
**Rating:** 5
**Confidence:** 4

**Summary:**

This paper proposes a method to construct fully connected graphs for crystal materials, with two attention biases between node pairs proposed. From my understanding, the first attention bias ($\alpha$ ones) captures infinite interactions between two nodes, and the second one ($\beta$ ones) captures the local interactions in the summation form. After this, the traditional fully connected transformer architectures can be applied. The experiments show a good improvements beyond Matformer.

**Strengths:**

The strengths of this method are as follows.

- A reasonably good transformer network with high efficiency for crystal scalar property prediction
- Good presentation, easy to follow writing
- Good efficiency for a transformer architecture

**Weaknesses:**

However, there are some concerns of this paper.

- One of the most important issue is that there is another work named PotNet[1], already published one year ago, proposed a method to calculate infinite summations for node pairs in crystal structures. The core idea in this paper of calculating the first attention bias ($\alpha$ ones) capturing infinite interactions between two nodes of this paper are very similar to PotNet. But the similarity is not mentioned, and comparison is not provided. Besides the first attention bias ($\alpha$ ones), the second attention bias ($\beta$ ones) is not new. It can be seen as the summation of local radius features.

- The derivation in the appendix might be problematic, the authors use word "may be", which is ambiguous. If authors want to use Equation (S14) in the Appendix to derive the first attention bias, which is one of their major contribution, the coefficient in the reciprocal space needs to be properly derived.

- The paper uses “computationally tractable” in many places, but the error bound of the summation is still unknown to me. Particularly, the authors say if the $\sigma$ of $\sum_n e^{-\frac{\Vert r\Vert^2}{2\sigma}}$ or $\sum_n \Vert r \Vert e^{-\frac{\Vert r\Vert^2}{2\sigma}}$ has an upper bound and lower bound then is tractable. If it is tractable, then what is the exact error between the authors’ evaluation and the value of every summation $\sum_n e^{-\frac{\Vert r\Vert^2}{2\sigma}}$

- The performances of the proposed method is not very significant without the additional stochastic weight averaging (SWA) (Izmailov et al., 2018) by averaging the model weights for the last 50 epochs with a fixed learning rate. If with this module, the comparison is not that fair, but without this module, the performance gains are not very significant.


[1] Efficient Approximations of Complete Interatomic Potentials for Crystal Property Prediction. (Public in ICLR 22 and accepted in ICML 23) https://proceedings.mlr.press/v202/lin23m.html

**Questions:**

The major questions are provided in the above weakness section. Given the concerns for now, I vote for rejection.

---

> ### Author Response · Authors · 2023-11-20
> **Responses to the concern about PotNet (1/2)**
>
> To Reviewer 8AQw:
>
> Thank you for your comments. We first respond to the following comment on PotNet, and then respond to other comments.
>
> ---
> > One of the most important issue is that there is another work named PotNet[1], already published one year ago, proposed a method to calculate infinite summations for node pairs in crystal structures. The core idea in this paper of calculating the first attention bias ($\alpha$ ones) capturing infinite interactions between two nodes of this paper are very similar to PotNet. But the similarity is not mentioned, and comparison is not provided. Besides the first attention bias ($\alpha$ ones), the second attention bias ($\beta$ ones) is not new. It can be seen as the summation of local radius features.
>
> Thank you for pointing out a relevant work, PotNet, by Lin et al. (2023). After a careful assessment of this work, we consider that our contributions remain sufficient even with the presence of PotNet for the following reasons.
>
> First, **their PotNet and our Crystalformer are built upon different architectural paradigms, GNN and Transformer.** While there are many GNN variants for crystal structure encoding, Transformer variants that have been successfully applied to crystal structure encoding are, to the best of our knowledge, limited to Matformer (Yan et al. 2022) and ours. Furthermore, as explained in Section 4, the Matformer architecture significantly differs from the standard Transformer architecture (Vaswani et al., 2017), for example, in the change of the softmax function to sigmoid and in the definitions of query/key/value features (see Figure 5 of the Matformer paper for more details). Although Matformer makes an important first step in introducing Transformers to the field of material modeling, we believe that **this field still lacks a viable Transformer baseline with minimal modifications to the standard Transformer architecture, except for our work.** In the field of ML, improvements and extensions of Transformer models have been and will be accumulated, mostly by assuming the standard architecture. Thus, **in order to continuously import these advances in ML into the field of material modeling, it is vital to have such a baseline Transformer model in the field.** This point is independent of the existence of PotNet.
>
> Second, **establishing a viable Transformer model for crystal structures is not trivial even when PotNet is known, because Transformers differ significantly from GNNs in their architectures and are generally more difficult to train.** For example, Transformer models usually require an initial warm-up phase with a low learning rate during training. Without this phase, the training typically fails to reach valid solutions. For our model, the presence of infinite summations makes the training even harder, and this widely used warm-up strategy could not lead to training success. As described in Section 3.2, **we found that the normalization-free Transformer architecture with an improved weight initialization strategy by Huang et al. (2020) is essential for a Transformer to successfully work for crystal structures**. Furthermore, **we have identified an architectural factor that enables proper modeling of the crystal structures of single unit-cell atoms** (see “Value position encoding for periodicity-aware modeling” in Section 3.1). As this modeling problem stems from softmax’s weight normalization, it is usually not recognized by GNNs. **These non-trivial findings will help future studies in this area** and are beyond the focus of the PotNet work.
>
> [*Remark: The third discussion below is better explained in a subsequent post:* https://openreview.net/forum?id=fxQiecl9HB&noteId=JqMYCMjipP *where we discuss that our method deeply fuses potential summantion calculations with neural networks by treating the whole attention operation as a neural interatomic potential summation.*]
>
> Third, **although PotNet uses the infinite summation of potentials similar to our $\alpha_{ij}$, its usage and definition are clearly different from ours**, as discussed below.
>
> ### PotNet's usage vs ours
> Specifically, PotNet computes infinite summations for each unit-cell atom pair, as $S_{ij} = \sum_{n} \Phi(r_{ij(n)})$, where $\Phi(r_{ij(n)})$ is a known physical interatomic potential function. Then, transform scalar $S_{ij}$ to a feature vector via radial basis functions (RBF) and multi-layer perceptrons (MLP), as $e_{ij} = \text{MLP}(\text{RBF}(S_{ij}))$. Then, feed it to a message passing function $x_i \gets g(x_i, \sum_j \text{f}(x_i, x_j, e_{ij}))$.
>
> In our case, we compute infinite summation $\exp(\alpha_{ij}) = \sum_n \exp(\phi_{ij(n)})$, which is similar to $S_{ij}$ in PotNet. However, it appears as part of softmax-based weighted averaging of features (i.e., attention): $\sum_n \exp(\phi_{ij(n)}) (v_j + \psi_{ij(n)})$ in Eq (3). Here, each potential value $\exp(\phi_{ij(n)})$ acts as a weight for feature vector $(v_j + \psi_{ij(n)})$.

---

> ### Author Response · Authors · 2023-11-20
> **Responses to the concern about PotNet (2/2)**
>
> [CONTINUED FROM "PotNet's usage vs ours"]
>
> Therefore, PotNet and our method use infinite summations quite differently. **In short, PotNet uses the infinitely summed potential values as abstract edge features fed to message passing functions, whereas we directly treat each potential value as an attention weight for weighted averaging of features.** In this regard, we also respectfully disagree with Reviewer 8AQw's comment that $\beta_{ij}$ "is not new. It can be seen as the summation of local radius features." Our $\beta_{ij}$ is formulated as $\frac{1}{Z_{ij}} \sum_n \exp(\phi_{ij(n)}) \psi_{ij(n)}$, essentially computing softmax attention (i.e., weighted averaging) of $\psi_{ij(n)}$ with potentials $\exp(\phi_{ij(n)})$ as weights. PotNet never uses potential values as weights for feature averaging.
>
> ### PotNet's definition vs ous
> PotNet defines $S_{ij}$ as a linear combination of different potential functions. Specifically, it computes three versions of $S_{ij}$ by setting $\Phi(r)$ to Coulomb potential $\epsilon_1/r$, London potential $\epsilon_2/r^6$, and Pauli potential $e^{-\alpha r}$, and computes their sum: $S_{ij} = S_{ij}^\text{coulomb} + S_{ij}^\text{london} + S_{ij}^\text{pauli}$. Also notice that $S_{ij}$ is a non-learnable function parameterized by three hyperparameters $\epsilon_1, \epsilon_2, \alpha$.
>
> We use Gaussian (and possibly other) potential functions with different tail-lengths for the parallel heads of multi-head attention. The tail lengths are learned along with the whole network, making the potential functions learnable. The parallel attention results with different potential functions are aggregated through subsequent neural network layers (i.e., "Linear" and "Feed forward" in the right part of Fig 3), which help to express more complex potential functions than the original forms, as discussed in Sec 6.
>
> In short, **PotNet combines different potential functions by their linear summation in potential space, whereas we abstractly fuse results with different potential functions in feature space through the multi-head attention mechanism of Transformer.**
>
> ---
> **Overall Summary**: Unlike PotNet based on GNN, we propose a Transformer-based model for crystal structure encoding. This difference leads us to a fundamentally different architecture from PotNet (even in the usage of infinite summation), with non-trivial findings specific to Transformers. Given the situation that there is a quite limited number of Transformer models for crystal structures, our work brings values that are irreplaceable by the PotNet work and that  will benefit the field of material modeling. In an upcoming revision, we will cite and discuss PotNet in the paper.

---

> ### Author Response · Authors · 2023-11-20
> **Point-by-point responses to other comments by Reviewer 8AQw (1/1)**
>
> Below we provide point-by-point responses to all other remained comments by Reviewer 8AQw.
>
> ---
> > The derivation in the appendix might be problematic, the authors use word "may be", which is ambiguous. If authors want to use Equation (S14) in the Appendix to derive the first attention bias, which is one of their major contribution, the coefficient in the reciprocal space needs to be properly derived.
>
> Sorry for causing the confusion by our misleading writing. Equation (S14) is derived concretely without ambiguity. Thus, “may be” is now rephrased by “is” in the paper.
>
> ---
> > The paper uses “computationally tractable” in many places, but the error bound of the summation is still unknown to me. Particularly, the authors say if the $\sigma$ of $\sum_n e^{-\frac{\Vert r\Vert^2}{2\sigma}}$ or $\sum_n \Vert r \Vert e^{-\frac{\Vert r\Vert^2}{2\sigma}}$ has an upper bound and lower bound then is tractable. If it is tractable, then what is the exact error between the authors’ evaluation and the value of every summation $\sum_n e^{-\frac{\Vert r\Vert^2}{2\sigma}}$.
>
> Thank you for pointing this out. **We have provided the error bound in the Supplementary Material (Section of “Bound for approximation error”)**.
>
> In summary, given the theoretical expression of the summation as
>
> $Z = \sum_{n} \exp \left(-\frac{\\| p_j + Ln - p_i \\|^2}{2\sigma_i^2} \right),$
>
> we approximate it as
>
> $\tilde{Z} =\sum_{n_1=-R_1}^{R_1}\sum_{n_2=-R_2}^{R_2}\sum_{n_3=-R_3}^{R_3} \exp \left(-\frac{\\| p_j+Ln-p_i \\|^2}{2\sigma_i^2} \right).$
>
> Here, as explained in the implementation details (Sec 3.5), ranges $R_1,R_2,R_3 \in \mathbb{Z_+}$ are adaptively determined to cover a spherical radius of $c\sigma_i$ in 3D space  (we emprically set $c=3.5$), while ensuring $R_1,R_2,R_3 \ge 2$.
>
> Then, residual error $\epsilon$ between $Z$ and $\tilde{Z}$ has the following upper bound:
>
> $\epsilon := Z - \tilde{Z} \le \frac{3}{2} \left( \frac{2}{\rho} \right)^3 \Gamma \left( \frac{3}{2}, \left( R-\frac{\rho}{2} \right)^2 \right)$
>
> where $\Gamma(z,x) = \int_x^\infty t^{z-1}e^{-t} dt$, $\rho = \min \\{ \frac{1}{\sqrt{2}\sigma_i} \\|Ln \\| \\, \vert n \in \mathbb{Z}^d, n \neq 0\\}$, $R = \max(c/\sqrt{2}, R')$, and $R' = \min \\{ \frac{1}{\sqrt{2}\sigma_i} \\| p_j + Ln - p_i \\| \\, \vert n \in \mathbb{Z}^3 , \max(|n_1|,|n_2|,|n_3|) > 2\\}$.
>
>
> ---
> > The performances of the proposed method is not very significant without the additional stochastic weight averaging (SWA) (Izmailov et al., 2018) by averaging the model weights for the last 50 epochs with a fixed learning rate. If with this module, the comparison is not that fair, but without this module, the performance gains are not very significant.
>
> Thank you for sharing your concern. As mentioned in our first response, **we aim to develop a new Transformer framework for crystal structure encoding, with equal emphasis on its effectiveness as well as conceptual and architectural simplicity.** This is because we consider the field of material modeling ($\neq$ molecular modeling) still lacks an effective baseline Transformer model with minimal modifications to the standard Transformer architecture. In this objective, we have presented our model as simpmle as possible in our framework, leaving room for further improvements as discussed in Sec 6. We believe that the results without SWA successfully demonstrate the effectivenss of this simple method.
>
> ---
> Thank you for all of these comments, which are helpful to brush-up our work. We are preparing a reivision to include PotNet.

---

> > ### Comment · Reviewer_8AQw · 2023-11-20
> > **Responses to authors**
> >
> > Dear authors,
> >
> > Thank you for these rebuttals, however, the justification provided is not very convincing because
> >
> > (1) The performance of PotNet is better than the proposed method according to PotNet paper, even with the SWA. And the performance gap is not small.
> >
> > (2) According to your title, Infinitely Connected Attention for Periodic Structure Encoding, the core difficulty is capturing infinite interactions between atom pairs, and as you said, this difficulty is tackled by previous works. The claim that you are just using it slightly different to fit into the simple Transformer design is not novel enough.
> >
> > (3) The alpha and beta, although with your further justifications, capture similar geometric information as previous method PotNet.
> >
> > Thus, I tend to remain the score of 5.

---

> ### Author Response · Authors · 2023-11-20
> **Additional responses to Reviewer 8AQw**
>
> We appreciate Reviewer 8AQw's participation in the discussion.
>
> > (1) The performance of PotNet is better than the proposed method according to PotNet paper, even with the SWA. And the performance gap is not small.
>
> Tables R1-R3 below show performance and efficiency comparisons with PotNet. We observe that our method is rather competitive to PotNet. This result is satisfactory when considering the following facts:
> - PotNet fully exploits the forms of three known potential functions (i.e., Coulumb $\epsilon_1/r$, London $\epsilon_2/r^6$, and Pauli $e^{-\alpha r}$) to take advantage of a strong inductive bias, while our method only uses simple Gaussian functions $e^{-r^2/2\sigma^2}$ (to serve as a missing Transformer baseline to the field of material modeling).
> - Our model requires only 47\% number of parameters of PotNet. (Table R3)
> - Our inference speed is 2.4x faster than PotNet. (Table R3)
>
> Despite our faster inference, PotNet is faster in training. This is because PotNet precomputes the infinite summations of the non-learnable potential functions for all training data, once at the beginning of training. By contrast, we employ learnable potential functions (i.e., Gaussian with learnable $\sigma$) and compute infinite summations from scratch at every training step. Despite this difference, our training time per epoch is competitive to PotNet.
>
> Here, we note that our latest implementation using an efficient softmax attention code runs much faster (Table R3, bottom) with statistically comparable accuracies. With this version, our method is faster than PotNet in training time per epoch.
>
> ---
> ### Table R1: Materials Project
> ||Form|BG|Bulk|Shear|
> |-|-|-|-|-|
> |PotNet|**0.0188**|0.204|0.040|**0.065**|
> |Ours w/o SWA|0.0198|**0.201**|**0.0399**|0.0692|
> ---
> ### Table R2: JARVIS-DFT
> ||Form|Total|BG (OPT)|BG (MBJ)|E hull|
> |-|-|-|-|-|-|
> |PotNet|**0.0294**|**0.127**|**0.032**|**0.27**|0.055|
> |Ours w/o SWA|0.0319|0.131|0.0342|0.275|**0.0482**|
> ---
> ### Table R3: Efficiency
> ||Train/epoch|Total train|Inference/material|# Params|
> |-|-|-|-|-|
> |PotNet |42 s|5.8 h|16 ms |1.8 M|
> |Ours|45 s|9.9 h|6.6 ms|853 K|
> |Ours (new code)|34 s|7.5 h|TBA| 853 K|
> ---
>
> > (2) According to your title, Infinitely Connected Attention for Periodic Structure Encoding, the core difficulty is capturing infinite interactions between atom pairs, and as you said, this difficulty is tackled by previous works. The claim that you are just using it slightly different to fit into the simple Transformer design is not novel enough.
> > (3) The alpha and beta, although with your further justifications, capture similar geometric information as previous method PotNet.
>
> "The claim that you are just using it slightly different to fit into the simple Transformer design" is not our claim, and alpha/beta represents a limited view of our method. (Sorry, maybe our explanation above was not clear enough.) There is a more fundamental (or conceptual) difference between PotNet and our method. Let's see how they perform state evolution.
>
> ### PotNet
> Using our notation, PotNet's state evolution is expressed as follows.
> $x_i' \gets g \left( x_i, \sum_j f(x_i, x_j, e_{ij}) \right)$
> Here, $f$ and $g$ are learnable neural network layers. Edge feature vector $e_{ij}$ is an *embedding* of an infinite summation of potentials, encoded via RBF and MLP as $e_{ij} = \text{MLP}(\text{RBF}(S_{ij}))$ where $S_{ij} = \sum_n \Phi(r_{ij(n)})$.
>
> Here, we see that **PotNet computes infinite summation $S_{ij}$ of potentials in a scalar potential space, transforms it to an abstract edge feature $e_{ij}$, and then the rest is done in a standard GNN framework with this new edge feature.**
>
> ### Ours
> Our state evolution, particularly its attention part (Eq. 3) , is expressed as follows.
> $y_i = \frac{1}{Z_i} \sum_j \sum_n \exp(q_i^Tk_j/\sqrt{d_K} + \phi_{ij(n)}) (v_j + \psi_{ij(n)})$
> As we describe in Sec 3.1, **this attention operation, as a whole, is interpreted as an infinite summation of interatomic potentials defined and performed in an abstract feature space**. Maybe it is better called *neural interatomic potential summation*, because overall potentials $\exp(q_i^Tk_j/\sqrt{d_K} + \phi_{ij(n)})$ are defined partly abstractly with features (e.g., $q$ and $k$), and summed values $(v_j + \psi_{ij(n)})$ are also features that abstractly express atom's states such as electric charges.
>
> **Summary**
> - PotNet exploits the values of infinite summations of existing potentials to derive new physics-informed edge features for GNNs.
> - Our method deeply fuses potential summation calculations with neural networks, by treating the whole attention operation as a neural interatomic potential summation performed in an abstract feature space.
>
> This **conceptual novelty of our work** should be taken into the rating.
>
> Also, the title could be modified to "Crystalformer: infinitely connected attention as neural potential summation for periodic structure encoding", etc., although not allowed during rebuttal.

---

> ### Author Response · Authors · 2023-11-22
> **Paper revised to reflect Reviewer 8AQw's comments**
>
> Dear Reviewer 8AQw,
>
> We have revised our paper to reflect the above discussions. The updates are summarized below and highlighted by blue color texts in the paper.
>
> - Updates related to PotNet
>   - We have added a discussion on PotNet in **Related Work (Sec 4)** and a more in-depth discussion in Section of **"Difference from PotNet" in the Supplementary Material**.
>   - We have added performance and efficiency comparisons with PotNet in **Tables 1, 2, and 3** as well as dicussions on these results in **Sec 5.2 and 5.3**.
> - We have further revised Introduction (Sec 1) and Method (Sec 3.1) to **more clarify the concept of the proposed method** by  regarding the infinitely connected attention as *neural potential summation*. (Although we had a similar explanation previously in Sec 3.1, we have more sharpened it so that the conceptual difference from PotNet gets clearer.)
>
> ## Contributions
> Lastly, we summarize our contributions again to wrap-up the discussions so far.
> - We present a simple yet effective Transformer framework for crystal structure encoding (Crystalformer) with **non-trivial architectural findings** (e.g., importance of value position encoding and normalization-free Transformer architecture).
> - We present computationaly-tractable **infinitely connected attention, with an intepretation as neural potential summation** that performs an infinite summation of interatomic potentials in abstract feature space.
> - Compared to Matformer
>   - Makes minimal modifications to the standard Transformer architecture and will serve as **a missing Transformer baseline in the field of material modeling**.
>   - Performs better, with 29.4% number of parameters of Matformer.
> - Compared to PotNet
>   - Different architectual paradims: Transformer vs GNN
>   - Different use of infinite summations (see also "Difference from PotNet" in Supplementary Material):
>     - Crystalformer **performs potential summations deeply in abstract feature space by the proposed attention.**
>     - PotNet computes potential summations as preprocessing to provide physics- and periodicity-informed edge features.
>   - Performs competitively
>     - with 48.6% number of parameters of PotNet
>     - without exploiting any known form of interatomic potentials (whereas PotNet fully exploits the forms of Coulomb, London dispersion, and Pauli repulsion potentials)
>
> Thank you very much for the discussions. With all the feedback, we believe that our paper has been sharpened and strengthened.

---

### Official Review · Reviewer_u628 · 2023-10-29

**Soundness:** 3 good
**Presentation:** 3 good
**Contribution:** 3 good
**Rating:** 8
**Confidence:** 4

**Summary:**

The paper proposes a new transformer architecture, called Crystalformer, for crystal structure property prediction that manages the periodic structure of crystals. To completely capture the periodic structure of crystals one would effectively require infinitely connected attention, which is intractable, prompting the papers to introduce a new attention method to effectively manage this requirement using a distance based decay.

The papers starts by introducing the problem of crystal structure property prediction and discusses recent approaches, including GNN-based neural networks and a couple of transformer models, specifically Matformer. The paper then introduces a set of preliminaries, including description of crystal structures and the self-attention mechanism. Subsequently, the authors describe the different components of Crystalformer with a focus on the attention formulation called pseudo-finite periodic attention that encodes relevant crystal information. Next, the authors describe distance decay attention, which applies a Gaussian decay function to place less importance on atoms that are further apart from each other and thereby simplifies the need for infinitely connected attention to a tractable problem. The authors describe relevant details of their attention formulation and distance decay function before outlining their network architecture consisting of multiple attention blocks.

In Section 4, the paper described related work in detail and emphasizes how Crystalformer differs from various competing approaches and showcase experiments on Materials Project and JARVIS in Section 5. The experiments in Section 5 show that Crystalformer generally outperforms relevant GNN and transformer baselines for both datasets. Next, the paper provided a set of ablation studies related to removing positional encodings and performing attention in Fourier space.

**Strengths:**

The paper has the following strengths:
* The paper presents a new attention formulation for crystal structure design that makes an infinite attention scenario computationally tractable (originality, significance).
* The paper provides detailed problem descriptions and related work outlining how their work differentiates itself from prior methods (clarity, significance).
* The paper provides relevant experimental results and ablations to supports its primary claims (quality).

**Weaknesses:**

The paper currently provides a somewhat limited scope in its experiments, especially for the datasets, and could be also improved by including additional relevant datasets and models:
* Datasets: OQMD [1], NOMAD [2] which are crystal structure datasets available through the Open MatSci ML Toolkit [3]. Experiments on these datasets would significantly strengthen the claims of the paper since they include significantly more datapoints than the ones in the current experiments. The toolkit also includes a more updated version of Materials Project with ~150k crystal structures.
* Models: FAENet [4], Equiformer [5]. While new experiments would be ideal to strengthen the paper claims, I would at least like to a discussion of these models in related work.


Additionally, the current draft would benefit from additional clarity related to the method, which are described in the section below.

[1] Kirklin, S., Saal, J.E., Meredig, B., Thompson, A., Doak, J.W., Aykol, M., Rühl, S. and Wolverton, C., 2015. The Open Quantum Materials Database (OQMD): assessing the accuracy of DFT formation energies. npj Computational Materials, 1(1), pp.1-15.

[2] Draxl, C. and Scheffler, M., 2019. The NOMAD laboratory: from data sharing to artificial intelligence. Journal of Physics: Materials, 2(3), p.036001.

[3] Lee, Kin Long Kelvin, et al. "MatSciML: A Broad, Multi-Task Benchmark for Solid-State Materials Modeling." arXiv preprint arXiv:2309.05934 (2023).

[4] Duval, Alexandre Agm, et al. "Faenet: Frame averaging equivariant gnn for materials modeling." International Conference on Machine Learning. PMLR, 2023.

[5] Liao, Yi-Lun, and Tess Smidt. "Equiformer: Equivariant graph attention transformer for 3d atomistic graphs." arXiv preprint arXiv:2206.11990 (2022).

**Questions:**

The paper could be improved by addressing the following questions related mainly to clarity:
* An example consisting of input features (atom types and atom positions) would make things clearer showing how the data is exactly processed. Figure 3 shows that positions and lattice parameters go into the attention blocks, but their processing is not explained in Figure 2.
* Can you describe stochastic weight averaging? It's first mentioned in Table 1.
* What is the reason for choosing your datasets? It seems like the Materials Project version is significantly smaller than the latest version. It would be good to have more details.
* Did you try other decay functions? Why or why not?

---

> ### Author Response · Authors · 2023-11-17
> **Point-by-point responses to Reviewer u628 (1/2)**
>
> To Reviewer u628:
>
> Thank you for your constructive feedback on our manuscript. Below we respond to all the concerns and questions by Reviewer u628, and have updated the manuscript accordingly.
>
> ---
> > The paper currently provides a somewhat limited scope in its experiments, especially for the datasets, and could be also improved by including additional relevant datasets and models:
> > - Datasets: OQMD [1], NOMAD [2] which are crystal structure datasets available through the Open MatSci ML Toolkit [3]. Experiments on these datasets would significantly strengthen the claims of the paper since they include significantly more datapoints than the ones in the current experiments. The toolkit also includes a more updated version of Materials Project with ~150k crystal structures.
>
> We appreciate your advice on how we can further strengthen our work.
>
> We agree that the inclusion of experimental comparisons with large-scale datasets can enhance the value of our work. However, training the proposed method and other competitors on such large-scale datasets requires rich computational resources or a not-cheap financial budget for renting them through cloud GPU services. For now, we would like to leave such large-scale experiments as our future work.
>
> Meanwhile, training on large-scale datasets itself is often done as independent topics, especially when Transformer-based large models are considered. For our case, it naturally attracts interest whether our Crystalformer can benefit from the increased dataset size by increasing its model size. Therefore, **we would like to consider Reviewer u628’s suggestion as one of the interesting future directions, and we have mentioned it from Sec 5.5 with references to OQMD and NOMAD**.
>
> See also our response to the question “What is the reason for choosing your datasets?” below.
>
> ---
> > - Models: FAENet [4], Equiformer [5]. While new experiments would be ideal to strengthen the paper claims, I would at least like to a discussion of these models in related work.
>
> Thank you for pointing out relevant references.
> We see that both FAENet (GNN-based) and Equiformer (Transformer-based) focus on developing techniques for SE(3)/E(3)-*equivariant* encoders, while our work targets SE(3)/E(3)-*invariant* encoders. Also, although Equiformer is a Transformer variant like ours, it is presented for finite molecule structures only. Therefore, we consider that they are not direct competitors to ours.
>
> However, we think that **their techniques for SE(3)/E(3) equivariance are conceptually orthogonal to our framework, and can therefore be utilized when extending our model** to be SE(3)/E(3)-equivariant depending on the target task, or to incorporate angular/directional information into our model. **We have mentioned such a possibility in Sec 6 (Section of “Angular and directional information”)** with new references to FAENet and Equiformer. Also, **we have added a reference to Equiformer in Sec 1 (3rd paragraph)** as a Transformer-based molecule encoder, cited along with Graphormer (Ying et al., 2021).
>
> ---
> > The paper could be improved by addressing the following questions related mainly to clarity:
> > - An example consisting of input features (atom types and atom positions) would make things clearer showing how the data is exactly processed. Figure 3 shows that positions and lattice parameters go into the attention blocks, but their processing is not explained in Figure 2.
>
> Thank you for your valuable suggestion. Following your advice, **we have updated Figure 2** as follows.
> - We have added a sub-figure showing an example of a crystal structure.
> - We have clarified that the spatial information (ie, point coordinates $\mathcal{P}$ and lattice vectors $\ell_1,\ell_2,\ell_3$) is used by “periodic spatial encoding” and “periodic edge encoding”.
>
> ---
> > - Can you describe stochastic weight averaging? It's first mentioned in Table 1.
>
> Thanks for the question. In both the initial and latest manuscripts, **stochastic weight averaging (SWA) is described in the main text of Sec 5.2**: “For test and validation model selection, we use stochastic weight averaging (SWA) (Izmailov et al., 2018) by averaging the model weights for the last 50 epochs with a fixed learning rate.”
>
> We believe that SWA can suppress performance fluctuations due to model selection and can therefore help to more reliably analyze the performance in the ablation studies.
>
> ---
> CONTINUE (1/2)

---

> ### Author Response · Authors · 2023-11-17
> **Point-by-point responses to Reviewer u628 (2/2)**
>
> ---
> > - What is the reason for choosing your datasets? It seems like the Materials Project version is significantly smaller than the latest version. It would be good to have more details.
>
> The Materials Project dataset (MEGNet’s snapshot) has been frequently used in the literature of ML-based material property prediction, for example, by Chen et al. (2019), Louis et al. (2020), Choudhary & DeCost (2021), and Yan et al. (2022). We use this dataset along with the JARVIS-DFT (3D 2021) dataset for the following reasons.
>
> - We think our method is most relevant to Matformer by Yan et al. (2022), who use the Materials Project (MEGNet) and JARVIS-DFT (3D 2021) datasets for their evaluations. Because they publicly share train/validation/test data splits, it was straightforward to use the same datasets and splits for a fair comparison.
> - Historically, early works on ML-based material property prediction often omitted to share their exact datasets and splits, making it difficult for later studies to perform comparisons as more methods emerge in this area. (Because of this situation, some papers even cite the error rates of methods trained and tested on different snapshots of the Materials Project with different dataset sizes.) Choudhary & DeCost (2021) and Yan et al. (2022) have made great effort to perform fair comparisons of many methods by training and evaluating them using the same datasets and splits. We considered that using the same datasets and splits as theirs leads to increased reproducibility and transparency of our work.
>
> **Section 5.2 explains that we use the same datasets and splits with Choudhary & DeCost (2021) and Yan et al. (2022)**. Additionally, **we have clarified that we use these datasets by “following our most relevant work by Yan et al. (2022)” in Sec 5**.
>
> ---
> > - Did you try other decay functions? Why or why not?
>
> For now we use only the Gaussian decay function.
>
> In Sec 6, we discuss the possibility of using other functions that approximate real interatomic potential functions. However, we think that such extensions are more domain-specific, and therefore **we plan to try them in a follow-up work aimed at the audience with more interest in materials science** (e.g., for a journal on materials science and informatics).
>
> Besides, the Gaussian decay function has the following benefits:
> 1. It exponentially decays in real space.
> 2. It has the analytic solution in Fourier (reciprocal) space, as shown in Eq (S14) in the Supplementary Material.
> 3. This Fourier-transformed form is also a Gaussian decay function in reciprocal space with an easy-to-understand complementary relation with the real-space Gaussian function. (Specifically, when the real-space Gaussian function has the tail-length of $\sigma$ in 3D space, the reciprocal-space Gaussian function has the tail-length of $1/\sigma$ in spatial frequency.)
>
> For these reasons, we think that the Gaussian decay function is the best choice to demonstrate the core idea of our method.
>
> ---
> Thank you for all of your comments, which have helped us to further polish our paper.

---

> > ### Comment · Reviewer_u628 · 2023-11-18
> >
> > Thank you for your detailed response and paper updates. Based on this, I have some additional comments:
> >
> > - **Experiments & Datasets:** I think the point about choosing datasets for the most apt comparisons of methods is relevant and I would like to see that better highlighted in the paper. I suggest to add additional details in Section 5 (or Section 5.1) about choosing the experimental settings you have (consistency and compute restrictions). Additionally, I suggest including a discussion on how training can scale to a more ambitious case (probably in Section 6 or Section 7), which would involve additional datasets (the Open MatSci ML Toolkit [1] includes NOMAD & OQMD, but also others that can be used which make the collection of data stronger) and compute. You can highlight an estimate of the required need for both there as well.
> > - **Limitations:** I suggest adding a discussion on some potential limitations of the method as well. For example, can this method be applied to more complex systems that include surface and interacting materials (e.g. OpenCatalyst [2])? What might need to be modified to do so?
> >
> > [1] Lee, Kin Long Kelvin, et al. "MatSciML: A Broad, Multi-Task Benchmark for Solid-State Materials Modeling." arXiv preprint arXiv:2309.05934 (2023).
> >
> > [2] Chanussot, Lowik, et al. "Open catalyst 2020 (OC20) dataset and community challenges." Acs Catalysis 11.10 (2021): 6059-6072.

---

> ### Author Response · Authors · 2023-11-21
> **Additional responses to Reviewer u628**
>
> To Reviewer u628:
>
> Thank you for your comments. With a tight space limitation, we have made our best to reflect Reviewer u628's comments as follows.
>
> - We have **revised Sec 5 to include a remark "about choosing the experimental settings"**, as suggested. (Note that the subsection of "5.1 Datasets" has been merged with Sec 5 for a smoother transition.)
> - We have also mentioned **"a more ambitious case" using large models and large datasets in Sec 7**.
> - **Limitations are actually already discussed in Sec 6 with possible solutions.** But we have **more clarified them in the revision** by the changing section name, etc. Specifically, sentences like  "such formulations limit the expressive power", "Our current setting limits them to the Gaussian decay function", and "such a bound possibly overlooks the presence of long-range interactions" in each of the three paragraphs represent limitations of the presented method.
>
> Note that Table 5 "Effects of attention in the Fourier space on the JARVIS-DFT dataset." that is referred from Sec 6, has been moved in the Supplementary Material as Table S3, to earn space.

---

### Official Review · Reviewer_NG2X · 2023-10-31

**Soundness:** 3 good
**Presentation:** 2 fair
**Contribution:** 3 good
**Rating:** 8
**Confidence:** 3

**Summary:**

To encode infinitely repeating crystal structures, this paper introduces a distance-decay inductive bias into the attention mechanism to ensure tractability. The authors compare their method with five baseline methods on two datasets.

**Strengths:**

1. The proposed decayed attention naturally incorporates a physical inductive bias, making the attention mechanism for infinite structures computationally tractable.
2. Good experimental results, and each component of the proposed method is also validated through ablation experiments.

**Weaknesses:**

1. The necessity of $\psi_{ij}(n)$ is unclear. Although the authors give an example in Page 4,  it remains questionable whether it is appropriate to assume $\mathcal{X} = \{x_1\}$ when attempting to prove the necessity of $\psi_{ij}(n)$.
2. Lack of discussion on reflection and periodic transformations [1].
3. About writing
	- In Figure 2, the representation of the infinitely connected attention is not clear.
	- In Figure 3, the interaction of $X^t$ with $\mathcal{P}$ and $l_1, l_2, l_3$ requires a more detailed and explicit illustration, as they are somewhat hidden within the formulas, which can be challenging for readers to locate.
	- "With simple algebra, we can rewrite Eq. (3) as..." This rewriting should be included in the appendix somewhere, and please point to it.


[1] Artificial Intelligence for Science in Quantum, Atomistic, and Continuum Systems

=======================

I sincerely appreciate the authors for taking the time and effort to address my concerns. I would like to raise my rating to 8.

**Questions:**

None

---

> ### Author Response · Authors · 2023-11-14
> **Point-by-point responses to Reviewer NG2X (1/2)**
>
> To Reviewer NG2X:
>
> Thank you for your comments. Below we provide point-by-point responses to them. We believe that our responses resolve all of the Reviewer NG2X’s concerns.
>
> ---
> > The necessity of $\psi_{ij(n)}$ is unclear. Although the authors give an example in Page 4, it remains questionable whether it is appropriate to assume $\mathcal{X}=x_1$ when attempting to prove the necessity of $\psi_{ij(n)}$.
>
> Thank you for sharing your concern with us. We think that our current discussion on the necessity of $\psi_{ij(n)}$ is basically fine for the following reasons.
>
> First, as a common practice in mathematics, **it is sufficient to give a single counterexample when proving that a mathematical statement/equation is problematic**. In “Value position encoding for periodicity-aware modeling” in Sec 3.1, we point out that when $\mathcal{X} = \\{ x_1 \\}$, Eq (3) without $\psi_{ij(n)}$ is reduced to $y_1 = v_1$, which is insensible to input lattice vectors. This modeling flaw can be fixed by explicitly inserting $\psi_{ij(n)}$ into Eq (3). Mathematically speaking, this observation sufficiently shows the necessity of $\psi_{ij(n)}$. (We have added detailed mathematical explanations in the Supplementary Material, as explained later.)
>
> Second, **from a materials science perspective, the shown counterexample ($\mathcal{X} = \\{ x_1 \\}$) is called *monatomic structures* and represents an important class of existing materials rather than imaginary ones.** Well-known monatomic structures of the same atom include *graphite* and *diamond*. These materials have the single unit-cell atom of Carbon but have different lattice vectors and thus different crystal structures. These differences make graphite a soft substance used for pencil tops while making diamond the hardest substance in the world. Thus, $\psi_{ij(n)}$ is required from a materials science view too.
>
> Finally, **Table 4 shows that the inclusion of $\psi_{ij(n)}$ consistently leads to performance improvements in all the prediction tasks**, emprically showing the the necessity of $\psi_{ij(n)}$.
>
> We have added the Section of **“Necessity of value position encoding” in the Supplementary Material** that summarizes the above discussion, and added a reference to it from the aforementioned part of Sec 3.1.
>
> ---
> > Lack of discussion on reflection and periodic transformations [1].
>
> As requested, **we have added a thorough discussion on invariance properties of the proposed model for various transformations**, including the permutation of unit-cell atoms, rotation/reflection and translation of 3D coordinates (so-called SE(3) and E(3) invariance), and periodic transformation. The section of **“Invariance properties” in the Supplementary Material** provides the definitions and proofs of these invariance properties. We have mentioned it from Sec 4 ("Invariance encoders").
>
> ---
> > About writing
> > - In Figure 2, the representation of the infinitely connected attention is not clear.
> > - In Figure 3, the interaction of $X^t$ with $\mathcal{P}$ and $\ell_1,\ell_2,\ell_3$ requires a more detailed and explicit illustration, as they are somewhat hidden within the formulas, which can be challenging for readers to locate.
>
> Thank you for your comments.
>
> Figure 2 illustrates our attention operation of Eq (4) in a matrix/tensor-based computational diagram instead of the vector-based representation of Eq (4). We adopt this style because **papers on Transformer architectures often illustrate a QKV attention operation by such a diagram, and following a similar manner should promote the understanding of readers who are familiar with Transformer papers.** Specifically, input vectors $x_1,...,x_N$ are concatenated as matrix $X = [x_1,...,x_N]$ in the diagram, and query, key, value, and output vectors are likewise expressed as matrices $Q, K, V, Y$. In this manner, scaled-dot product $q_i^Tk_j/\sqrt{d_K}$ for all ij pairs is expressed as $Q^TK/\sqrt{d_K}$. Also, $\alpha_{ij}$ for all ij pairs is expressed as matrix $A$ to be added to $Q^TK/\sqrt{d_K}$. Similarly, $\beta_{ij}$ for all ij pairs is expressed as rank-3 tensor $B$ to be added to $V$.  These notations are shown in the diagram. After all, Figure 2 is an illustration of Eq (4) in a common computational diagram of QKV attention and presents no new information than Eq (4).
>
> However, we think Figures 2 and 3 had some clarity problems, which may have caused confusions. Specifically, how $\mathcal{P}$ and $\ell_1,\ell_2,\ell_3$ are used in Figure 2 was unclear, and how Figure 2 and 3 are related was unclear.
>
> Thus, **we have updated Figures 2 and 3** as follows.
> - For Figure 2, we have clarified that $\mathcal{P}$ and $\ell_1,\ell_2,\ell_3$ are used as input to “Periodic spatial encoding” ($\alpha_{ij}$) and “Periodic edge encoding” ($\beta_{ij}$), and further clarified that Figure is a “matrix-tensor diagram” in the caption.
> - For Figure 3, we have clarified that its “Multi-head attention” block corresponds to Figure 2.
>
> ---
> CONTINUED (1/2)

---

> ### Author Response · Authors · 2023-11-14
> **Point-by-point responses to Reviewer NG2X (2/2)**
>
> > "With simple algebra, we can rewrite Eq. (3) as..." This rewriting should be included in the appendix somewhere, and please point to it.
>
> Thank you for your suggestion. **In the Supplementary Material (Section of “Derivation of pseudo-finite periodic attention”), we have added a step-by-step derivation from Eq. (3) to Eq. (4), and pointed to it from the main text.**
> Note that in the initial manuscript, Eq. (4) had a mis-formulation: ${q_i^T k_{j}}$ had to be ${q_i^T k_{j}}/\sqrt{d_K}$. We apologize if it caused any confusion. This mis-formulation has been fixed too.
>
> ---
> Thank you again for all of these comments. With the new content to reflect Reviewer NG2X's comments, we believe that our paper has been strengthened.

---

### Official Review · Reviewer_S6vd · 2023-11-01

**Soundness:** 4 excellent
**Presentation:** 3 good
**Contribution:** 3 good
**Rating:** 8
**Confidence:** 4

**Summary:**

This paper presents an attention mechanism and a transformer model for periodic structure encoding. The attention mechanism is formulated as an infinite sum over the periodic extensions of the unit cell in all three directions. In order to make the sum tractable, the paper proposes a distance decay attention by which the spatial dependencies between atoms decay exponentially with their distance. The authors present an evaluation of this attention mechanism, incorporated into a transformer architecture, on two data sets and several material property prediction tasks, comparing with a handful of methods from the crystal modelling literature.

**Strengths:**

This paper is clearly written and easy to follow. I have enjoyed reading it and I believe it will likewise be relevant for the machine learning community working on materials modelling. Concretely, these are some of the strengths I see in this paper:

- The introduction effectively describes the gap in the literature it addresses, namely the periodic nature of crystal structures, which is not considered in the core of the GNN literature devoted to molecular modelling. The paper provides a comprehensive review of graph and transformer models for related tasks and motivates the need for a method to account for the periodic nature of crystal structures.
- Sections 2 and 3 clearly describe the proposed model and the figures effectively support the text descriptions.
- The evaluation seems sound and the results achieved by the proposed method are remarkable compared to existing approaches, including in terms of training and inference time.

**Weaknesses:**

I do not have any major concerns. If I were to put on the hat of the average machine learning reviewer, I could argue that the technical contribution of this paper is limited because the method is a small adjustment of existing work. However, that is largely irrelevant to me provided the idea is sound, it is clearly described, it addresses a current challenge in materials modelling and the results are competitive.

One comment regarding the presentation of results that I can make is that it would be much easier to interpret the results if they were presented graphically instead of in tables, despite this being the common practice in the machine learning community.

**Questions:**

- The results in Table 4 demonstrate that the inclusion of the value position encoding systematically improve the performance of the model. However, we observe significant differences in the performance gap between the proposed and simplified models across properties. Do you think this gap in performance could be explained by the characteristics of each property?
- I am also curious whether you have some intuitions about why the performance of the dual space version of the model is significantly better than the baseline in the case of predicting the energy above hull.

---

> ### Author Response · Authors · 2023-11-13
> **Point-by-point responses to the two questions by Reviewer S6vd**
>
> To Reviewer S6vd:
>
> Thank you for your insightful comments. In this research we aim to establish a new Transformer framework for crystal structure encoding, with equal emphasis on its effectiveness as well as conceptual and architectural simplicity, so that both the ML and materials science communities can easily understand/use it and flexibly extend it in future studies. We appreciate that Reviewer S6vd finds the values of our research as we hope, and we believe that there are more than a few people like Reviewer S6vd in the communities who wish to use/extend our work to further advance this field.
>
> ---
> > The results in Table 4 demonstrate that the inclusion of the value position encoding systematically improve the performance of the model. However, we observe significant differences in the performance gap between the proposed and simplified models across properties. Do you think this gap in performance could be explained by the characteristics of each property?
>
> Thank you for your question.
> As pointed out, Table 4 shows that the inclusion of the value position encoding yields greater improvements for the formation energy and total energy while relatively limited improvements for the bandgap and energy above hull (E hull). This result can be understood from the fact that **the formation energy and total energy are absolute energy metrics, whereas the bandgap and energy above hull are relative energy metrics that evaluate gaps between two energy levels**, as $\Delta E = E_1 - E_0$.
>
> The accurate prediction of such energy gaps is generally difficult, because it needs accurate predictions of both $E_1$ and $E_0$. Furthermore, even when an algorithmic improvement in the prediction method leads to improvements in absolute energy prediction (e.g., formation energy prediction), it does not necessarily lead to improvements in energy gap prediction. Specifically, assume that an improved algorithm predicts energies $E_1' = E_1 + \Delta_1$ and $E_0' = E_0 + \Delta_0$ as better predictions than $E_1$ and $E_0$, respectively. These improvements can directly lead to lower prediction errors of absolute energy metrics. However, when improvements $\Delta_1$ and $\Delta_0$ are made by the same mechanism, it may be possible that $\Delta_1 \simeq \Delta_0$ so that they are canceled out in energy gaps $\Delta E' = E_1' - E_0'$.
>
> We conjecture that the limited improvements in the bandgap and energy above hull prediction are due to these difficulties of energy gap prediction.
>
> ---
> > I am also curious whether you have some intuitions about why the performance of the dual space version of the model is significantly better than the baseline in the case of predicting the energy above hull.
>
> Thank you again for your question. Although we are not aware of the exact mechanism for this behavior, we conjecture that it is related to the fact that **the energy above hull is the only metric in Table 5 that involves the energy evaluation of different crystal structures than the targeted one**.
>
> A more in-depth discussion needs to know the definition of the energy above hull. Let’s again express it as $\Delta E = E_1 - E_0$, where $E_1$ corresponds to the formation energy of the target structure and $E_0$ is the so-called convex hull provided by stable phases of the target structure.
> As we know, substances around us change their phases between solid, liquid, and gas depending on the temperature and pressure. Likewise, a crystal structure has multiple stable phases with different structures within the solid state. For example, a structure with a chemical formula of $AB$ has phases that have different structures with chemical formulas of $A_{x}B_{1-x}$, $x \in [0,1]$ (i.e., structures with the same element composition but different ratios of elements).
> When we plot y = FormationEnergy($A_{x}B_{1-x}$) for all possible phase structures of AB, we can draw a convex hull of the plots, which will have some corner points that represent stable phases of AB. (One can obtain an intuitive illustration of this by searching for “energy convex hull”, such FIG. 5 in https://www.researchgate.net/figure/A-schematic-convex-hull-in-the-A-B-chemical-space-Phases-Si-lie-on-the-convex-hull-and_fig2_312191506, where S1 to S4 represent stable phases of AB.) The energy above hull is then defined as the vertical deviation between the formation energy of the target structure and the line of the convex hull, essentially representing the instability of the target structure.
>
> When $E_1$ and $E_0$ involve the energy evaluation of different structures, it may be possible that the inclusion of long-range interactions has a positive impact on some of the stable phases of a given structure in $E_0$ while not so for the given structure itself in $E_1$. This asymmetric relation between $E_1$ and $E_0$ may mitigate the aforementioned cancellation effect and may lead to improved prediction of $E_0$ and $\Delta E$.
>
> We will add these insights in an upcoming revision.

---

> > ### Author Response · Authors · 2023-11-21
> > **Revision made to include the discussions**
> >
> > To Reviewer S6vd:
> >
> > We have made a revision to include the above discussions. They are provided in the following parts.
> > - Section **"Physical interpretaions of results in Table 4" in the Supplementary Material**. Referred from Sec 5.4
> > - Section **"Results using the dual space attention" in the Supplementary Material**. Referred from Sec 6.
> >
> > Thanks.

---

> > > ### Comment · Reviewer_S6vd · 2023-11-22
> > > **Final comments**
> > >
> > > Dear authors, I have read your detailed responses to the reviewers and I appreciate the updates in the manuscript. I have no further questions about the submission.

---

### Author Response · Authors · 2023-11-22
**Uploaded a clean version of the paper**

Dear all Reviewers,

As the end of the rebuttal period is approaching, we have uploaded a clean version of our paper.
- **The main pdf now contains the main paper and all the appendices** (from A to J) in order of appearance.
- As the supplementary material, we have uploaded the same content but major changes are hilighted in red or blue.

Major updates from the initial manuscript are summarized as follows.
- Improved method explanations in Introduction and Sec 3.1 in terms of *neural potential summation*. (Reviewer 8AQw)
- Improved Figure 2. (Reviewers NG2X and u628)
- Addition of PotNet in Related Work (Sec 4) and Experiments (Sec 5.2 and 5.3). (Reviewer 8AQw)
- Detailed explanations about experiment settings in Sec 5. (Reviewer u628)
- New discussions on FAENet and Equiformer in Sec 6. (Reviewer u628)
- Appendix A: Derivation from Eq (3) to Eq (4).  (Reviewer NG2X)
- Appendix B: Error bound for the infinite summation calculation. (Reviewer 8AQw)
- Appendix D: Discussion on the neccessity of value position encoding. (Reviewer NG2X)
- Appendix F: Proofs of invariance properties. (Reviewer NG2X)
- Appendix G: Difference from PotNet. (Reviewer 8AQw)
- Appendix I and J.4: Physical interpretations of results.  (Reviewer S6vd)
- etc.

(See also a summary of contributions in https://openreview.net/forum?id=fxQiecl9HB&noteId=kSJxMAArIl)

We believe that this latest version addresses all the concerns raised by all the reviewers. Thank you for all the discussions.

---

### Meta-Review · Area_Chair_4JWf · 2023-12-08

**Metareview:**

The authors present a neural network architecture which can be applied to atomistic models of crystals to predict their properties. The model generalizes the idea of Ewalds summation to apply to self-attention layers in the hidden units of the network, enabling attention to be applied to an effectively infinite crystal, overcoming finite size effect limitations. The results are strong, the only sticking point with one reviewer being the comparison with PotNet. The authors have added an extensive discussion comparing with PotNet which I believe addresses most of the reviewer's concerns. One point in the authors' response which confused me was their insistence that PotNet somehow has an unfair advantage because it uses potentials motivated by known physics, but I fail to see how this is an issue. If possible, the paper could be made stronger by investigating the Crystalformer with the same potentials as PotNet as well as the current version. However, even without this, I still believe the paper is over the threshold for acceptance.

**Justification For Why Not Higher Score:**

This is already addressed in the metareview.

**Justification For Why Not Lower Score:**

This is already addressed in the metareview.

---

### Decision · Program_Chairs · 2024-01-16

Accept (poster)